# Locally Prepared Therapeutic Food for Treatment of Severely Underweight Children in Rural India: An Interventional Prospective Controlled Community-Based Study with Long Follow-Up:—‘SAMMAN’ Trial

**DOI:** 10.3390/nu16172872

**Published:** 2024-08-27

**Authors:** Ashish Rambhau Satav, Vibhawari S. Dani, Jayashri S. Pendharkar, Kavita Ashish Satav, Dhananjay Raje, Dipty Jain, Shubhada S. Khirwadkar, Eric A. F. Simões

**Affiliations:** 1MAHAN (Meditation, AIDS, Health, Addiction, Nutrition) Trust, Melghat, C/o Mahatma Gandhi Tribal Hospital, Karmgram, Utavali, Dharni, Amravati 444 702, India; drvsdani@gmail.com (V.S.D.); cao@mahantrust.org (J.S.P.); drsatav@rediffmail.com (K.A.S.); stats@mahantrust.org (D.R.); diptyjain57@gmail.com (D.J.); paeds@mahantrust.org (S.S.K.); 2Department of Pediatrics, Section of Infectious Diseases, University of Colorado School of Medicine, Aurora, CO 80045, USA; eric.simoes@cuanschutz.edu; 3Department of Epidemiology, Centre for Global Health, Colorado School of Public Health, Aurora, CO 80045, USA

**Keywords:** case fatality rate, community based, growth trajectory, India, long term follow-up, prevalence, recovery, relapse, rural, severe underweight (SUW)

## Abstract

Background: Severely underweight (SUW) children contribute significantly to under-five mortality and morbidity. There are WHO guidelines for the management of severe acute malnutrition but no specific guidelines for SUW management. Objective: The objectives were to achieve a recovery rate of 30% at 90 days of treatment for severe underweight (SUW) children aged 6–60 months, compare changes in weight-for-age Z (WAZ) scores, growth patterns, and case fatality rates between intervention and reference arms (RA), and reduce the prevalence of SUW in the intervention arm (IA). The target of a 30% recovery rate was achievable and significant based on our past research conducted in similar settings. Methods: Design: A prospective controlled community-based, longitudinal, two arms (IA, RA), intervention study with long follow-up was conducted between January 2011 and October 2023. Setting: Primary care for participants from 14 villages in rural Melghat, India. Participants: The study participants included SUW children aged 6–60 months and age-matched (±2 weeks) normal controls. The SAMMAN (Acronym for SAM-Management) intervention was comprised of local therapeutic food-micronutrient (LTF-MN) therapy for 90 days, intensive behavior change communication, infection treatment, and quarterly anthropometric records. SUW recovery, growth patterns, case fatality rate, prevalence at 90 days of therapy and at 60 months of age, and survival until early adolescence were assessed. ANCOVA analysis was used to obtain changes in Z-scores. Results: In the IA, the recovery rate was 36.8% at 90 days and 78.2% at 60 months of age. The mean difference in change in WAZ scores between the intervention arm and the reference arm was statistically significant (*p* < 0.0001). Growth patterns were similar between the two arms up to early adolescence. The SUW case fatality rate was significantly lower in the IA (0.9%) as compared to 4.62% in the RA at 60 months (*p* = 0.022). The reduction in SUW prevalence in intervention villages was higher than in the control villages (*p* < 0.001). The cost of management per SUW child was 3888 INR (47 USD) less than RUTF. Conclusion: The SAMMAN intervention is safe and cost-effective for significantly improving WAZ scores, sustainable, and hence replicable in resource-limited areas.

## 1. Introduction

Malnutrition contributes to 45–68.2% [1,2,3,4,5] of under-five mortality and, for survivors, has long-term implications for educational achievement, economic productivity, and the risk of noncommunicable diseases in later life [4]. The Millennium Development Goals had set a target of reducing the prevalence of malnutrition to half between 1990 and 2015, which was not achieved. The Sustainable Development Goals (SDG) target 2.2 is to end all forms of malnutrition by 2030 [6]. The UNICEF, WHO, World Bank Group joint malnutrition estimates focus on the prevalence of wasting (WHZ), stunting (HAZ), and obesity, and stress on the prevention of stunting [7]. They used mid-upper arm circumference and weight for height (WHZ) criteria for malnutrition assessment and treatment^5^ and did not use weight for age (WAZ) [7,8] to define or treat malnutrition. Moderate underweight babies (MUW) usually precede the more severe manifestations of acute chronic malnutrition, SUW, which is defined as WAZ less than three standard deviations (SD) below the median [9]. Use of the weight for height anthropometric criteria, misses a large swath of SUW, [10,11,12] eventually leading to the lack of development of potential solutions for its prevention and management. The prevalence of SUW is more than double that of severe acute malnutrition (SAM). Our published study (2016) has revealed the prevalence of SAM as 7.1% and severe underweight (SUW) as 18.7% [10]. The study by UNICEF (2018) has revealed the prevalence of SAM and SUW in tribal areas of India as 5.66% and 20.27%, respectively [12]. While the global prevalence of SAM was 3.9% for boys and 2.5% for girls, [11] and that of SUW was 9.8% (boys) and 7.3% (girls) (2017) [11].

The prevalence of SUW (WAZ < −3SD) [9] is 11% in India [13], overall 12.1–18.7% in rural India [14,15], and 18.7–36.1% in tribal India [10,12,16,17]. The pooled mortality hazard ratio (HR) of SUW is high, 4.6–9.40 [18,19,20,21,22,23,24,25,26].

The WHO guidelines for the Recommended Dietary Allowance (RDA) of macro- and micronutrients and duration of therapy for the management of severe acute malnutrition are available; however, there are no specific guidelines for SUW management. Their nutritional requirements, and length of therapy are unknown. The study by Islam highlights the importance of implementing the present treatment guidelines of SAM for facility-based management of severely underweight children [26].


What was already known:


PubMed, Cochrane, Google-Scholar, and B.M.J. databases were searched for global and Indian studies using the following terms—SUW, community–based management of severe malnutrition, ready-to-use therapeutic food (R.U.T.F.), LTF, etc., until March 24.

In almost all previous studies, feeding is unsupervised and supplementary in nature. The ready-to-use therapeutic food (R.U.T.F.) packets/ration are distributed as takeaways to be fed at home. However, this has not significantly impacted SUW recovery/mortality. 

‘MAHAN’ has been providing hospital and community-based health and nutrition services to the rural—tribal community of Melghat, India, since 1998.

We initiated this trial, because of the significant burden of SUW and the lack of guidelines for the community-based management of SUW. This study aims to assess the impact of community-based management of SUW using locally prepared, culturally acceptable, affordable, palatable multiple therapeutic foods with micronutrients on SUW children.

The primary objective is to achieve a recovery rate of approximately 30% at the end of 90 days and to assess recovery at the age of 60 months in severely underweight (SUW) children between the ages of 6 and 60, in a population of 14,000 from 14 tribal villages in Melghat, central India. The target of a 30% recovery rate was achievable and significant based on our past research conducted in similar settings [1]. The secondary objectives are: (a) To compare the change in WAZ score and growth pattern of the intervention arm and the reference arm (normal age and time-matched) up to the age of 5 years in the above setting. (b) To compare the case fatality rate of SUW children in the Intervention Arm (IA) with the non-SUW in the Reference Arm (RA), at three stages: at the end of 90 days of treatment, at 60 months of age, and at early adolescence (9–13 years). (c) To reduce the point prevalence of SUW by at least 35% at the age of 5 years (2019) as compared to baseline (2011) in intervention villages. The exploratory objective was to assess the impact of SUW management on the SUW relapse rate up to the age of 60 months.

## 2. Methods

### 2.1. Study Design

A prospective controlled community–based, longitudinal, two arms (intervention arm and reference arm), intervention study was conducted with long term follow-up, from September 2010 to October 2023.

### 2.2. Setting and Participants

Melghat, a rural-tribal/area in Maharashtra, India, consisting of 320 villages (clusters), divided into Dharni and Chikhaldara blocks, with a population of approximately 300,000, of which 84% are tribals. The Dharni block was divided into five zones, after stratification, based on distance from base hospital, three clusters from each zone were selected. One cluster dropped out due to an unwillingness to participate. Thus, 14 representative study villages were randomly selected (Appendix A). These villages represent the demographic, socioeconomic characteristics, health indicators, and malnutrition (SUW) of maximum Melghat villages. 

The study participants included the ‘intervention arm’ which consisted of SUW children aged 6–60 months, and the ‘reference arm’, which consisted of age-matched (±2 weeks) control children with normal WAZ (non-SUW), from the same demographic study area, whose parents gave informed written consent and were permanent residents of the study area. Children who migrated from the study area for more than six months were excluded from the study.

The purpose of adding a comparative reference arm of normal children was to know whether the impact of our intervention on recovery, growth pattern, BMI, and case fatality rate of SUW children is similar to, non-SUW normal children at the age of 60 months. Thus, the treated SUW children in the intervention arm would have similar outcome indicators to those of normal children in the reference arm. Our ethical committee did not permit, diagnosing SUW children and yet not treating them, as that would be unethical. Hence, we included normal, non-SUW children as RA. 

The children in the intervention arm and reference arm were selected from the same 14 study villages. Thus, it was not possible to calculate the impact on prevalence due to our intervention. Thus, to compare and analyze the impact of our interventions on SUW prevalence in the 14 study villages, another group of 22 control villages was randomly selected. 

We conducted our intervention in this study at the village level, and therefore those villages are named as the intervention villages. In the intervention villages, we targeted severely underweight children (SUW) as the intervention arm for treatment by LTF-MN to know its impact. In the same intervention villages, we selected reference arm non-SUW children, age-matched (±2 weeks), as controls.

The Independent Institutional Ethical Committee of MAHAN approved the study on 1 April 2011. Ethical code number 1.

The trial registration number is ClinicalTrials.gov. Identifier: NCT 02671786.

### 2.3. Phases of Study

The preparatory phase was from January 2011 to April 2011. During this phase, we conducted microplanning, questionnaire design, team building, obtained community consent, and completed a sociodemographic survey (Appendix A).

We appointed the field workers, VHWs, supervisors (medical, data collection, and behavior change communication), and retrospective surveyors. The VHWs were local, tribal, married, semiliterate, and socially sensitive women selected through community meetings.

Intervention phase was conducted from May 2011 to May 2015.

(A)Enrollment: Every month, VHWs conducted a door-to-door survey and recorded the anthropometry of all children in the villages between 6 and 60 months of age using a complete enumeration method, standardized digital weighing machines, and stadiometer/infantometer.

Our team of experts selected all SUW from the anthropometry data. Our trained medical supervisors cross-checked anthropometry of all SUW and SAM children, other supervisors cross-checked anthropometry of MUW and MAM. The final list of SUW children was prepared for enrollment in interventions. Thus, during enrollment, we included all the SUW children present in the villages. SUW children were enrolled after passing the appetite test as per WHO guidelines (Appendix A). Those with complications or who failed the appetite test were referred to the hospital. Those who refused hospitalization were also enrolled after their parents’ high–risk consent. Weekly anthropometry was conducted during LTF-MN therapy.

(B.1)SAMMAN intervention was conducted in the intervention arm. It has a duration of 90 days for each SUW child.
a.Local Therapeutic Food and Micro-Nutrients (LTF–MN): The enrolled SUW children received, LTF-MN feedings, four times a day, under direct supervision of VHW at feeding center in the village. 

Our LTF consisted of 8 different varieties of culturally acceptable ready-to-eat and ready-to-heat feeds. The ready-to-eat feeds like ‘chivda’ and ‘chikki’ were given as breakfast and evening snacks, and ready-to-heat feeds like ‘khichadi’,’thalipeeth’, and ‘upama’ were given for lunch and dinner after cooking at home for VHWs. From 1998 to 2010, we conducted a diet survey, multiple interviews, and focus group discussions to understand the dietary pattern and cultural acceptability of food by the local community, especially the children. We designed our LTF as per the dietary pattern and cultural acceptability of local children. 

We also used the WHO guidelines, specifically for the management of ‘Severe Acute malnutrition’ (SAM) [27], in the form of Recommended Dietary Allowance (RDA) for macronutrients and micronutrients, while designing LTF.

Our LTF (100 g)-MN (5 g) provides 13.5–16 grams of proteins, 450–550 calories, and 20–30 g of fats, along with 40 micronutrients as per WHO UNICEF guidelines [28].

Breast-feeding was allowed and promoted. Thus, each child received four to six grams of protein/kg/day and 175 kcal/kg/day with gradual escalation. Micronutrient deficiency was corrected by giving specially designed MN powder supplement, five grams per 100 g of LTF, which provided all essential micronutrients as per WHO-UNICEF guidelines. After 90 days, the child received home-cooked food (Appendix A).

    b.Our VHWs provided treatment for associated infections under the guidance of medical supervisors and provided appropriate referrals. Standard guidelines, as per WHO protocols [29,30] were followed (Appendix A).    c.Behavior Change Communication: Weekly intensive behavior change communication was conducted for the parents in the intervention arm during 90 days of therapy. Hand washing, nail cutting, and other personal hygiene were ensured (Appendix A).    d.Our VHWs daily examined all SUWs for medical and other complications, and provided appropriate childcare.

(B.2)In the reference arm, “Standard care”, consisting of behavior change communication once in 10 weeks and treatment of infections, as per WHO protocols [29,30], was provided. VHWs provided referrals to government hospitals for the needy children. The children from the reference arm who became SUW after enrollment were referred to government hospitals for malnutrition management.(C)After the SAMMAN Intervention of 90 days, “standard care”, was provided for children in both arms until the age of 60 months.

Follow-up after Intervention (till May 2019): Quarterly follow-up was conducted until the age of 60 months for anthropometry and mortality in the intervention arm and reference arm; relapses in intervention arm/conversion to SUW in the reference arm. 

Follow-up for survival and BMI was conducted until early adolescence (until October 2023) in both arms. Our village health workers (VHWs) recorded the anthropometry of all study participants in the villages, every month, until the age of 60 months (2011–2019). In 2019, the age group of study participants was from 4 to 9 years. All study participants were annually followed up during home visits by VHWs until the adolescent age. The anthropometry of all the study participants was recorded once after 5 years, i.e., in 2024, by VHWs. The age group of study participants in 2024 was 9–14 years, i.e., adolescence age. All the data was verified by supervisors.

(D)The prevalence of S.U.W. was determined in 14 intervention villages and 22 control villages in every September, from 2011 to 2022.(E)Costing: Expenses incurred for LTF-MN ingredients, transport, manpower, and the cooking process were considered for cost calculation.(F)Operational definitions:

SUW [8,27,31] is defined as WAZ ≤ –3SD; MUW: WAZ > –3SD and < –2SD of the median; and Normal: WAZ > –2SD of the median, following WHO Growth Reference data 2006 [8,27,31].

Outcome indicators are recovery, growth pattern, case fatality rate, prevalence, relapse, and body mass index (BMI) (Appendix A).

## 3. Statistical Methods

### 3.1. Sample Size

The primary objective was to achieve a recovery rate of at least 30% in SUW children after 90 days of SAMMAN intervention. Based on our pilot study, we assumed that, to obtain this proportion with a 95% confidence level, and 5% tolerable error, a sample of 323 SUW children would be required. Further, assuming a 5% loss to follow-up, a total sample of 339 SUW children was decided. Equivalently, 339 age-matched non-SUW normal WAZ children were also included to understand the growth trajectories and prevalence of SUW in two groups during follow-up until early adolescence.

The calculation of the wealth index was based on a categorical principal component analysis of household assets. The L3norm score was obtained from three components of major variability. The scores in the highest quartiles indicate good wealth status. The comparison from baseline to 3 months and baseline to 60 months of age in the intervention arm and reference arm was performed using analysis of covariance (ANCOVA).

Observations with more than 80% missing values during the follow-up were omitted from the analysis. The LOCF method was referred to for any missing anthropometric measurements. For the intervention arm, dichotomous outcomes recovered/not–recovered’ and ‘recovery–with- relapses/recovery–without relapse’ were considered. Multiple logistic regression was used to see the outcome at 90 days of therapy and 60 months of age. The village–level categorization was determined by referring to Wald’s statistics (Appendix A).

All the analyses were performed using R–4.4.1 (R Foundation for Statistical Computing, Vienna, Austria) package, and the statistical significance was evaluated at a 5% level.

### 3.2. Patient and Public Involvement (PPI)

From 1998 to 2010, a collaborative research initiative involving tribal communities, parents, and other stakeholders aimed to address the major concern of SUW through interviews, focus group discussions (F.G.D.), and community engagement. This identified the high prevalence and mortality of SUW and the low health-seeking behavior of the community. The present study integrated PPI, utilizing gramsabhas (community meetings), F.G.D.s, interviews, and door-to-door surveys (Appendix A).

PPI was an important component of the success of the trial. Without PPI, it was not possible to achieve the targets.

## 4. Results

The study includes a CONSORT flow chart of the participants. The deaths from both arms were ignored in the analysis, while loss-to-follow-up cases were retained in the downstream analyses, as seen in Figure 1.

Table 1 describes the socioeconomic and demographic characteristics (age of enrollment, sex, birth weight, parent literacy, type of community, wealth index from assets, and village level status) of children in intervention arm and reference arm villages. The distribution of characteristics, except birth weight, did not differ significantly between the two arms.

Recovery of SUW children (WAZ > −3) after 90 days of therapy was 36.8%, which improved by 41.4% (95% CI: 34.2%, 48.4%) and became 78.2% at the age of 60 months. The proportion of non recovered SUW children from the intervention arm (14.7%) was less than the proportion of converted SUW children from non-SUW children in the reference arm (19.2%), at the age of 60 months. (*p* = 0.152), as shown in Figure 2.

A comparison of WAZ scores was conducted in the intervention arm and the reference arm. The estimated marginal means for change in z–scores in the two groups, after adjusting for age at enrollment, and fixed factors like gender, birth weight, parent literacy, and wealth index, as shown in Table 2, were calculated. The mean change in the intervention arm from baseline to 3 months, was positive, suggesting an improvement in the health status and being significantly better than the reference arm (*p* < 0.05). Estimated marginal means for change in z–scores from baseline to 60 months of age in the intervention arm had a positive mean change and significantly higher than the reference arm (*p* < 0.05). The mean differences for the baseline to 60 months were higher than the baseline to 3 months. The profiles of raw mean WAZ scores in two arms at baseline, 3 months, and 60 months of age of children according to age at baseline are given in Figure 3. Improvement was observed in the intervention arm across time in all the age categories as compared to the reference arm.

The overall mean change in the intervention arm is: 0.76 ± 0.17 and in the reference arm: −0.12 ± 0.20 after adjustment with covariates. The mean difference between the intervention arm and reference arm is 0.88 ± 0.18 (0.58, 1.34), and it was statistically significant with a *p* < 0.0001 (Table 2). The unadjusted mean change in the z–scores for the intervention arm was 0.84 (SD: 0.80), while that for the reference arm was –0.6 (SD: 0.88) across all age groups (Figure 4).

A multiple logistic regression analysis revealed that parental literacy and birth weight (≥2500 g) have a significant positive impact on recovery at 3 months as well as 60 months. Moreover, the improvement in village-level characteristics significantly reduced the risk of recovery with relapse (*p* < 0.05), as shown in Table 3. Other factors showed a statistically nonsignificant effect on the risk of outcomes at 3 months and 60 months of age.

### Recovery with/without Relapse and Associated Factors

Recovery without relapse was treated as a reference outcome, while recovery with relapse was considered as the outcome of interest. Out of the 265 recovered children, 138 (40.7%) had no relapse and were normal at the end of the study. Younger children, in the age group of 6–24 months, had significantly more relapses, i.e., 61% (81) compared to 35% (46) in the 25–60 months age group (*p* < 0.0001). Significant risk factors for relapses were scarce communication facilities, the difficulty of accessing healthcare, and the absence of safe drinking water. A comparison between recovery at 60 months with and without relapse showed that the likelihood of recovery without relapse increased significantly as village-level facilities improved. 

The case fatality rate was calculated as the number of SUW deaths per 100 SUW children during the specified time period (Appendix A). The case fatality rate of all participants was 0.29% in the intervention arm, which was less than the case fatality rate in the reference arm (0.59%) during SAMMAN intervention period. In the reference arm, 65 non-SUW children became SUW until the age of 60 months. The case fatality rate of these converted SUWs in the reference arm was 4.62%. The case fatality rate of SUW in the intervention arm was 0.9%. The difference (−3.73% [95% CI: −8.92, 1.46]) was significant (*p* = 0.022). The difference in the proportion of mortality among all participants between the intervention arm and the reference arm, from the age of 60 months until adolescence was insignificant (*p* = 0.999).

The annual point prevalences were calculated after door-to-door anthropometric survey by MAHAN team every September from 2010 to 2023. There is a significant reduction in S.U.W. prevalence in I.A. by 14.04% (95% CI: 11.9%, 17.4%) (*p* < 0.001). In 24 control villages from another set, the prevalence was reduced by 2.4% (95% CI: 0.8%, 3.9%). The difference in prevalence between intervention and control villages was 11.64% (95% CI: 9.86%, 13.42%) (*p* < 0.001) (Figure 5).

Follow-up was conducted until early adolescence (October 2023), to understand the long-term impact on survival and anthropometry. All children were successfully tracked for survival. Sixty-five children from the intervention arm and 142 children from CA went to other villages; hence anthropometry could not be performed. There was no difference in case fatality rates in both groups (1.17%). The difference in mean BMI and the proportion of mortality between the two groups was not significant (*p* = 0.952 and *p* = 0.999, respectively). The probability density of BMI followed a normal distribution in both groups, as revealed in Figure 6. The follow-up of treated SUW children revealed that they achieved growth patterns and BMIs similar to those of normal adolescents, thus, indicating a sustained positive impact of our intervention.

The alluvial plot displays the transition of SUW children as regards severe stunting and severe wasting from baseline to 60 months of age. The recovery in severe wasting and severe stunting was observed in 86.7% and 45.3% cases, respectively, at 60 months of age, as per Figure 7.

In the intervention arm, there was a significant improvement in the normal/M.A.M. children at the end of 90 days of treatment and also at 60 months of age (*p* < 0.0001). The reduction in the prevalence of SAM as well as severe stunting from baseline to the end of 60 months of age was significant (*p* < 0.0001), as shown in Appendix A.

The cost per SUW child in the present study was INR 3888 (47 USD) for LTF.

LTF has better recovery rate, less case fatality rate, less drop out, is cost-effective than RUTF. Hence, LTF is sustainable and replicable, as per Appendix A.

Harm: Refeeding diarrhea was noted in 6 (1.79%) SUW cases after LTF, and 8 (2.39%) SUW had vomiting after mineral mix.

## 5. Discussion

Study design: A prospective, longitudinal, controlled, community–based study was implemented through VHWs in the difficult-to-access rural-tribal Melghat region of India. SAMMAN intervention of 90 days duration was given to 339 SUW children aged 6–60 months. The intervention included LTF-MN nutrition therapy, treatment of infections, intensive B.C.C., and daily childcare and hygiene. 

Follow-up of children was conducted in the intervention arm until the age of 60 months for recovery, change in WAZ scores, growth pattern, case fatality rate, and prevalence of SUW. Follow-up for their survival and BMI was conducted until early adolescence.

The WHO guidelines, for the management of ‘Severe Acute malnutrition’, (SAM) [27,29], regarding Recommended Dietary Allowance (RDA) for macronutrients, micronutrients and duration of therapy, are available. However, similar guidelines for ‘acute on chronic malnutrition’(SUW), are not available. In the absence of specific guidelines, we used the available WHO guidelines [23,29], for SUW children from 2011 to 2023. Our LTF-MN (100 g and 5 g, respectively), provides 13.5–17 g of proteins, 450–550 calories, and 20–30 g fats, with micronutrients as per WHO guidelines [27]. The government of India has started a similar program in 2023 [32].

The recovery rate for SUW was 36.8% after 90 days of SAMMAN intervention, which increased to 78.2% at 60 months of age. Age-wise WAZ scores of the intervention arm showed significant improvement over the reference arm until 60 months of age (*p* < 0.001). They catch-up with the normal growth velocity curves of the age matched controls by 60 months of age. This underlines the importance of timely and appropriate community-based treatment of common infections and behavior change communication until 60 months of age, implemented by VHWs.

The SUW recovery rate was 28.3% in a pooled data analysis of nine countries [22]. In Hossain’s study in Bangladesh, the rates of weight gain were greater in the SUW group who received cereal based supplementary food (*p* < 0.05) as compared to other groups who did not receive supplementary food [18].

The ComPAS trial (group-3) had 16.7% recovery with RUTF with 48.4% loss to follow-up [23].

The Udupi RCT study, [2] with monthly intensive behavior change communication for 12 months, without food supplementation, had a recovery rate of 20.5% for uncomplicated underweight children (severe and moderate) of 3–5 years of age. 

The recovery rate of SUW children was not significant in the intervention group of children who were provided supplementary food in Iran (*p* = 0.62) [33]. In the community nutrition project of Senegal [34], the recovery rate of children with WAZ < −3 was 18.2%.

In the Rural Malawi RCT study [35], corn-soya blends or lipid-based nutrients were monthly distributed for 12 weeks at home for SAM children of 6–18 months age. Their secondary data analysis revealed insignificant recovery of SUW (*p* = 0.211) because of the short duration of unsupervised supplementary feedings, the lack of intensive behavior change communication, and the lack of monitoring for sharing food with other family members. 

In the Bangladesh Integrated Nutrition Project, supplementary food was given to severely underweight children in the community. The recovery of SUW was not significant, nearly zero impact on WAZ [36]. The possible causes are mistargeting, sharing of supplementary food; substitution for other foods, inadequate food, incomplete participation, culturally less palatable food, no intensive behavior change communication, no capacity building activities, and inadequate feeding practices [37].

Most studies had primarily targeted SAM management and had conducted a secondary analysis of SUW [22,23,38]. The possible causes of low recovery in SUW in most other studies may be supplementary and unsupervised feeding, lack of field monitoring, short duration of treatment, no behavior change communication, and no community-based treatment of infections, and no long-term follow-up. The caregivers were rarely adequately instructed about feeding at home [22,38]. The criteria for recovery was not WAZ but MUAC > 12.5 cm [22,23,39].

Factors for recovery were parental literacy and a normal birth weight. SUW cases with low birth weight (LBW) showed significantly poorer recovery at the end of 90 days of therapy (*p* = 0.049). The children born undernourished, are less likely to recover within 90 days. However, by 60 months, there is no significant difference between the WAZ score of LBW and normal birth–weight children. The SUW children born as LBW can achieve normal growth by 5 years, if they are provided with LTF-MN nutritional therapy and intensive weekly behavior change communication for 90 days, followed by routine behavior change communication and management of childhood illnesses till the age of 60 months, irrespective of socioeconomic conditions (Appendix A).

The sustained impact of this study is evident by the fact that at the age of 60 months, the proportion of children with SUW in the intervention arm (14.7%) is less than that in the reference arm (19.2%). In the natural course of growth, some of the children who were normal in the beginning in the reference arm became SUW due to common infections like pneumonia, diarrhea, malaria, tuberculosis, etc. Other factors were poverty and the chronic scarcity of nutrition. In the intervention arm, due to LTF-MN therapy, treatment of infections, and intensive behavior change communication, the proportion of children with SUW was less than in the reference arm.

Sustained impact: Intensive behavior change communication improved the behavior of parents regarding childcare, nutrition, and health-seeking, which was sustained up–to the age of 60 months. LTF, which was prepared in accordance with WHO norms, the dietary needs and tastes of SUW children, was readily accepted and approved by the parents. Post 90 days of LTF-MN therapy, when home-cooked food could be started for kids, parents started cooking food in a similar manner, a direct result of intensive behavior change communication over weeks, as seen by the change in WAZ scores. (Figure 3). While RUTF was considered as medicine sachets meant only for sick babies, LTF was accepted as food. Most other studies provided unsupervised supplementary feeding or take–home rations without adequate instructions, intensive behavior change communication, and community-based treatment of infections [22,38,39].

### 5.1. Recovery with/without Relapse and Associated Factors

The recovery without relapse was treated as a reference outcome, while recovery with relapse was considered as the outcome of interest. Out of the 265 recovered children, 138 (40.7%) had no relapse and were normal at the end of study indicating importance of our intensive behavior change communication and treatment of infections till the age of 5 years. Younger children, in the age group of 6–24 months, had significantly more relapses i.e., 61% (81) compared to 35% (46) in the 25–60 months age group (*p* < 0.0001). During early age, the immunological system is week as compared to older children and hence more prone for infection. During the growth trajectory, we also noticed the episodes of infection (pneumonia, diarrhea, tuberculosis) which preceded the relapses.

There is an increase in masticatory performance (chewing capacity) of children as they grow up [40,41]. Similarly, the chewing capacity of smaller children is less as compared to older children, hence the relapses are more in early age group. The mature rotatory chewing movements of molar teeth and backward pushing of food by tongue comes during the age of 24 to 30 months which help for proper intake and digestion of food in older children [42]. Behavior change communication should focus on cooking of separate special food for children below the age of 24 months by parents, after completion of LTF therapy. 

Significant risk factors for relapses were scarce communication facilities, difficult to access healthcare and the absence of safe drinking water. Comparison between recovery at 60 months with and without relapse showed, that the likelihood of recovery without relapse increases significantly as the village-level facilities improved. 

Few studies have treated SUW, without focusing on relapse, default rate, infections [2,33], and recovery [43]. There were fewer adverse effects (4.18%) in present study as compared to the Malawi study, having 17.4% adverse effects during the intervention period [35]. All adverse outcome (combination of deaths, defaulters, nonresponse, transfer) were 37.9% in SUW in pooled data of 9 countries [22].

The case fatality rate in intervention arm (0.29%) was less than that of reference arm (0.59%) during SAMMAN intervention period, which is significantly less than WHO guidelines for SAM (<4%). This is the major achievement of a present study as most other SUW studies had higher case fatality rate of SAM and SUW as compared to normal children [21,22,37,44]. The case fatality rate of SUW was 1.8% in pooled analysis of 9 countries [22], and 1.88% in Malawi [35], 1.7% in ComPAS trial [23], 0.98% in Hossain’s study [18] and 1.2% in Bangladesh hospital [26]. Mortality reduction post-SAMMAN was sustained up-to early adolescence. Most of other studies treated uncomplicated SAM and SUW, while we treated both uncomplicated and complicated SUW and SAM.

The children in the reference arm who were initially normal but eventually became SUW were offered standard care and treatment through government facilities. 

Prevalence: The significant reduction in the prevalence of SUW in intervention villages (*p* < 0.001) (Figure 5) indicates the sustained impact of SAMMAN intervention. The LTF-MN therapy of 90 days, intensive behavior change, communication, and treatment of infections augmented the recovery of SUW and prevented the relapse. In intervention villages, treatment of infections and behavior change and communication among all children also prevented occurrence of new SUW cases. Intensive behavior change communication changed child rearing, child feeding, and hygienic practices of parents. It resulted not only in reduction in the prevalence of SUW but also improved the growth pattern and achieved a BMI similar to normal children.

Impact of intensive behavior change communication without supplementary feeding:

In the Palghar study, the prevalence of SUW was reduced from 32.9% to 23.9% (*p* value = 0.002) due to intensive behavior change communication for 18 months by health staff without food supplementation [3]. In the Udupi study, the prevalence of SUW reduced significantly in the intervention area (from 8.69% to 3.16%) as compared to control area because of health education and nutrition demonstrations [2]. In our previous study of tribal areas [44], the prevalence of severe malnutrition was reduced significantly, net reduction by 50.52% in intervention villages over control villages (*p* < 0.01) with intensive behavior change communication and community based treatment of infections without supplementary feeding [45].

Supplementary feeding with behavior change communication:

In the tribal Rajasthan study, the supplementary feedings containing a total of 500–700 calories per day were given as three supervised feeds in government Nutrition Care Centers and two feeds at home. Follow-up was conducted for 6 months. There was no intensive behavior change, communication, and treatment of infections. Hence, the SUW prevalence in the Rajasthan study was reduced from 32.9% to 26.1% [43].

Supplementary feeding without behavior changes communication:

In a Bangladesh study, supervised supplementary feeding (8–9 g proteins, 300 kcal) was given to severely underweight children for 4 months, with no significant difference in the prevalence in the project and nonproject areas (11.4% vs. 12.1%) because of improper and inadequate supplementary food that replaced home food, inadequate counseling of parents, reduced or no appetite of sick SUW, and sharing of food with other siblings [46].

The government of India accepts the reduction in prevalence as a major criterion to evaluate the impact of any nutrition intervention. Hence, we studied the point prevalence of SUW in intervention villages and compared it with the above studies. 


The Interest of using LTF-MN: Appendix A.


The cost of SUW child management was I 3888 NR/child (47 USD) for 90 days of LTF–MN therapy, which is cheaper than the RUTF of the Maharashtra Government, INR 9438/child (113 USD) [47] and ComPAS study (standard protocol 1041 USD) [23]. A systematic review by Fetriyuna has revealed that local therapeutic foods are more cost effective than standard RUTF [48]. The low cost of LTF-MN used in our intervention is attributed to using locally available food products and preparations by local manpower with the use of minimal technology. On the other hand, RUTF is industrially produced with a higher marketing cost. LTF-MN being cheaper, we could provide LTF to a large number of SUW children and SAM children from 2011–2023. Nutrition programs with LTF-MN therapy can be implemented on a larger scale, without increasing the economic burden by the government and other organizations, which will help to achieve the SDG Goal 2.2.

LTF is prepared locally, creating employment for tribal females [49]. It is prepared in a socioculturally accepted way, with a variety of preparations, that add diversity and palatability to the food, overcoming the limiting factors. LTF builds confidence in the local community that severe malnutrition can be treated with their own food [1]. Hence, the acceptance rate is >93% [1]. RUTF is a single dish, and hence the acceptance rate is very low (40%) [50]. A systematic review by Fetriyuna has revealed that local therapeutic foods are more acceptable than standard RUTF [48].

LTF is perceived as food prepared by local females, and it helped in improving cooking practices in the community even after therapy due to behavior change communication. RUTF is considered as industrial and medicinal sachet for sick children and not food. It will not improve cooking practices in the community. Hence, people using RUTF lose confidence in home food for SUW treatment.

In most studies, RUTF packets for 1 to 2 weeks are given to parents, to be consumed at home by severely malnourished child; however, sharing with other members of family cannot be controlled, which affects the recovery of the index case [51]. In the present study, all LTF-MN feeds given three to four times a day, are directly supervised by VHW at the community feeding centre, in clean and hygienic surrounding. This has improved our recovery rate. Hence, therapeutic food should be given to SAM, SUW in nutrition feeding centers, not at the homes of the beneficiaries. MAHAN LTF-MN has a better recovery rate for SUW as compared to RUTF (<30%) [22,23].

The case fatality rate after RUTF is 1–5%, [22,23,52], even though most of these studies have treated uncomplicated SUW. In our study, the case fatality rate is 0.29% for SUW who were treated with LTF, even when, we have treated both complicated and uncomplicated SUW.

Our prior study of LTF-MN therapy for SAM management had a recovery rate of 79.4% [1].

In the present study, analysis of SAM and severe stunting out of treated SUW in the intervention arm showed significant improvement in the recovery and reduction in proportion of SAM (*p* < 0.0001), and severe stunting (*p* < 0.0001), indicating the impact of SAMMAN intervention on most serious cases of combined SAM and SUW, and severe stunting and SUW. This highlights that the recovery of stunting is possible with LTF-MN, during long term follow-up coupled with regular behavior change, communication, and treatment of infections.

Thus, LTF is more sustainable and replicable as compared to RUTF.

The BMI of the intervention arm and the reference arm children were almost the same up–to early adolescence, indicating that the recovered SUW children follow a growth pattern similar to normal children. This sustained impact is possible due to intensive behavior change communication, which positively changed the child rearing, feeding, and hygienic practices of parents; and timely and appropriate community-based treatment of infections. We did not come across any study with such long-term follow-up [2,18,19,22,23,33,38,53,54,55] (Appendix A).

### 5.2. Value Addition of This Study

This study adds to a strategy for community-based management of SUW. The recovery rate has increased to 78.2% at the age of 60 months after 90 days of nutritional therapy. This multipronged approach covers (a) supervised LTF feeding, (b) behavior change communication, and (c) community-based treatment, resulting in significantly improved recovery and, reduced SUW mortality and prevalence (*p* < 0.001).

## 6. Conclusions

This study, with a good recovery, few adverse effects, and culturally acceptable, safe, and cost-effective with locally available resources, can be a sustainable and replicable model for other impoverished, difficult-to-access settings globally. Nutrition programs should include WAZ < –3 as an additional independent criterion for malnutrition therapy [23,56].

### Limitations

The study design could not be R.C.T. as it was unethical to refuse therapeutic food to a diagnosed SUW in the reference arm. This is a single-center study. LTF preparations need to be tailored according to the local flavors and tastes.

The strength of this study is long-term follow-up (a) until the age of 60 months for recovery, mortality, prevalence, and relapses, and (b) until early adolescence for survival. We could not find studies with follow-up until 60 months [2,18,19,22,23,33,38,53,54,55]. The SAMMAN intervention has significantly lower mortality than the WHO protocol for SAM management and other SUW studies [22,26,57]. The key features are a strong PPI, very few adverse effects, and cost-effectiveness.

## Figures and Tables

**Figure 1 nutrients-16-02872-f001:**
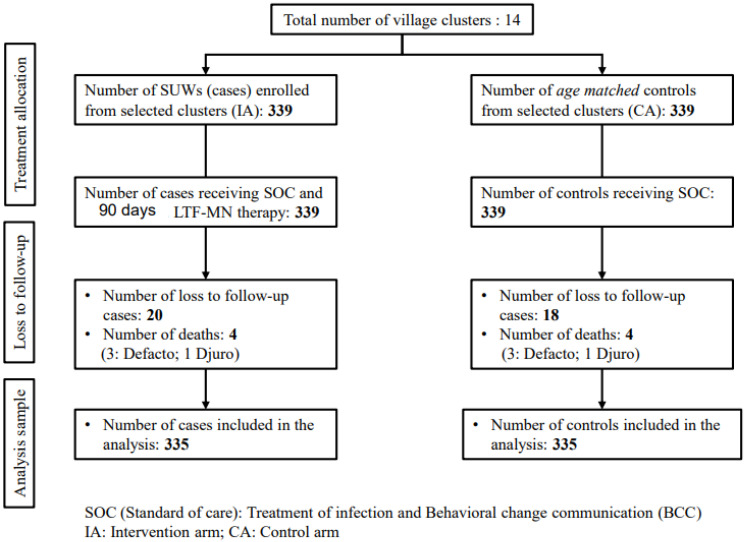
CONSORT Flow chart. (control/reference arm).

**Figure 2 nutrients-16-02872-f002:**
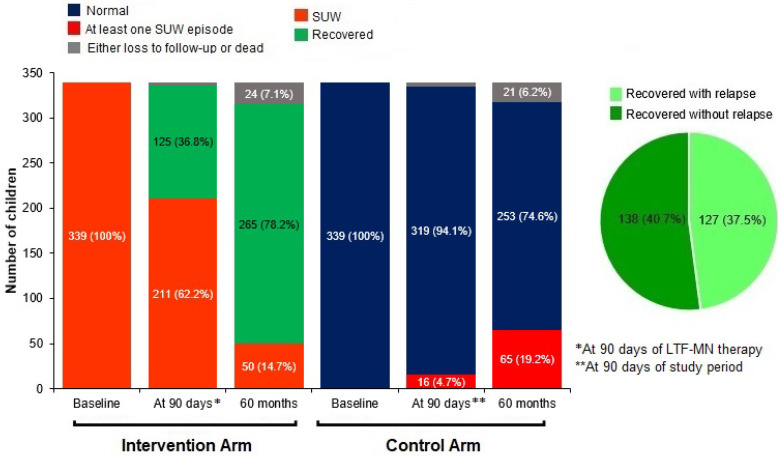
Recovery of SUW in intervention arm and nutritional status in reference/control arm.

**Figure 3 nutrients-16-02872-f003:**
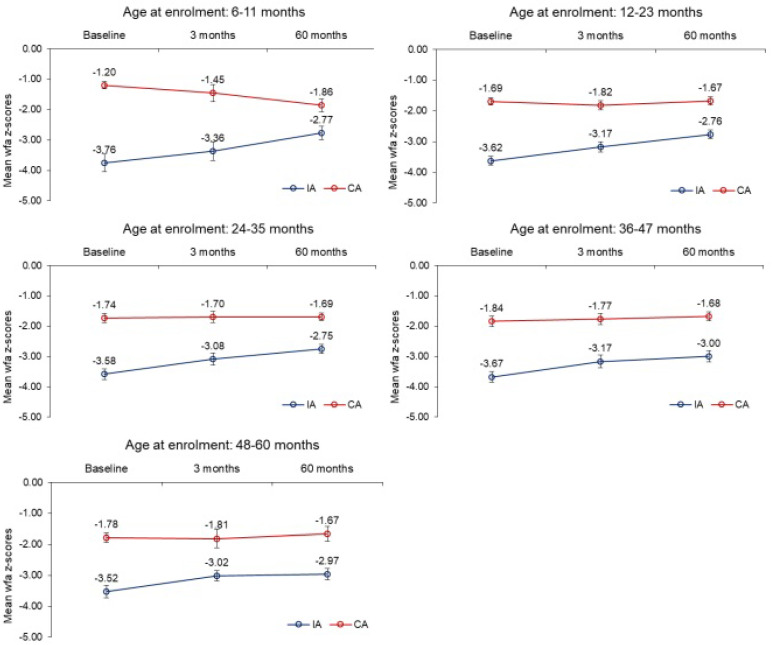
WAZ score profiles in the intervention arm and reference arm.

**Figure 4 nutrients-16-02872-f004:**
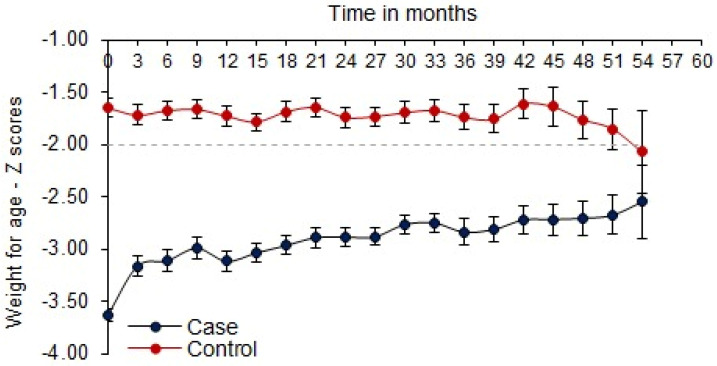
WAZ trend in intervention arm and reference arm during 2011–2015.

**Figure 5 nutrients-16-02872-f005:**
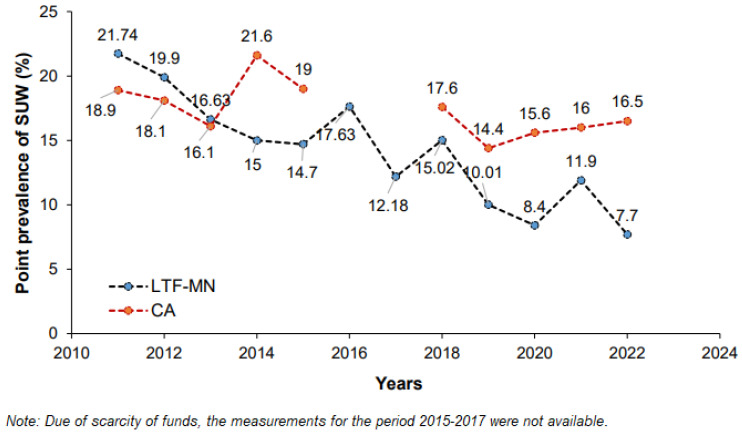
Point prevalence across different years. (LTF-MN is for intervention villages, CA-control villages).

**Figure 6 nutrients-16-02872-f006:**
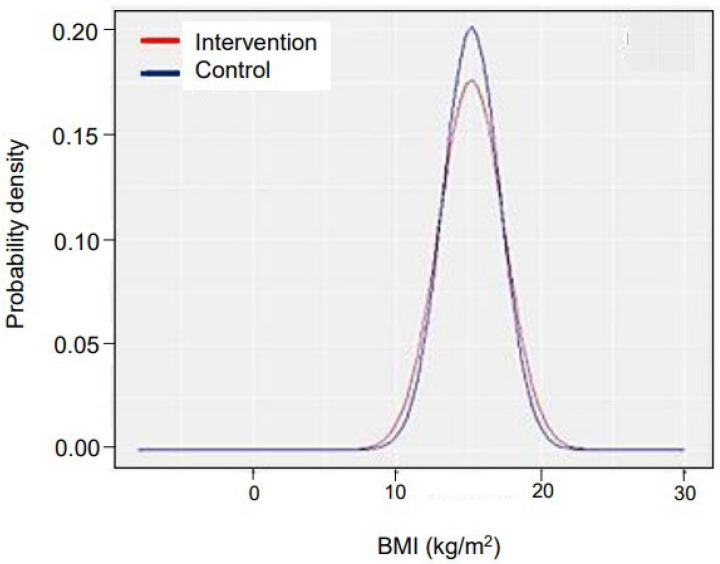
BMI of children from 5 to 12 years of age in both arms.

**Figure 7 nutrients-16-02872-f007:**
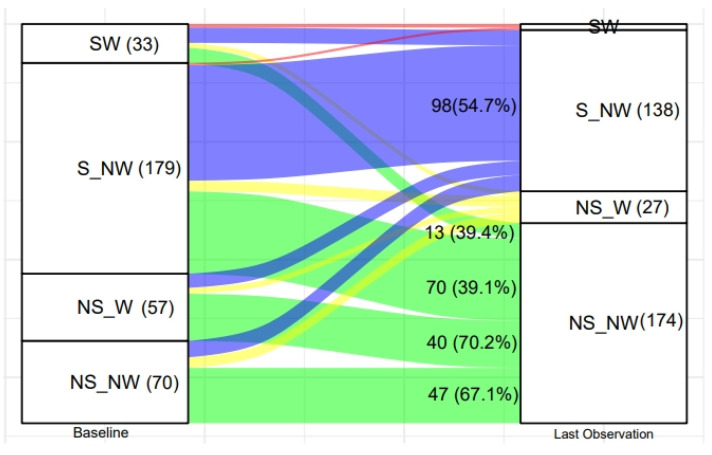
The alluvial plot displaying the transition of SUW children as regards severe stunting (S) and severe wasting (W) from baseline to 60 months of age. (NW: not severely wasted, NS: not severely stunted).

**Table 1 nutrients-16-02872-t001:** Distribution of children according to demographic and socioeconomic factors in intervention and control arms.

Factors	Levels	Arm		*p*-Value
		Intervention (N = 339)	Control (N = 339)	
		N (%)		
Age at enrolment (months)	6–11	54 (15.9)	58 (17.1)	0.991
	12–23	112 (33.0)	114 (33.6)	
	24–35	88 (26.0)	84 (24.8)	
	36–47	50 (14.7)	49 (14.5)	
	48–60	35 (10.3)	34 (10.0)	
Sex	Female	161 (47.5)	171 (50.4)	0.442
	Male	178 (52.5)	168 (49.6)	
Birth weight (g)	<2500	177 (52.2)	3 (0.9)	<0.001
	≥2500	162 (47.8)	336 (99.1)	
Parent literacy	Both illiterate	34 (10.0)	26 (7.7)	0.279
	Either parent literate *	305 (90.0)	313 (92.3)	
Community	Nontribal	20 (5.9)	16 (4.7)	0.493
	Tribal	319 (94.1)	323 (95.3)	
Wealth index from assets ^1^ **	I	97 (28.6)	74 (21.9)	0.185
	II	78 (23.0)	89 (26.3)	
	III	85 (25.1)	83 (24.6)	
	IV	79 (23.3)	92 (27.2)	
Village level status ^2^ **	I	128 (37.8)	118 (34.8)	0.281
	II	78 (23.0)	78 (23.0)	
	III	38 (11.2)	28 (8.3)	
	IV	95 (28.0)	115 (33.9)	

^1^ Parameters: Land ownership, House walls, Roof top, Electricity, House ownership, Toilet facility, Fuel, Kitchen, Smoke outlet, Windows/room, Drinking water source, Four-wheeler, Two-wheeler, Color TV, Bicycle, Wrist-watch, Radio, Newspaper, Mobile phone, Fan, Bullock cart, Cow, Buffalo, Bullock, Sheep, Cock, Hen, Winter precaution, Woolen clothes, Wood fire. ^2^ Parameters: Health Sub-center (HSC), Distance from HSC, Distance from base hospital, Distance from SDH, Road facility, Road facility in rainy season, Mobile/TV connectivity, Private Road transport, Government Road transport, Emergency health service, Village health workers, number of workers, ANM workers, ASHA workers, Anganwadi workers, Gram panchayat, Public distribution system. * Who can read, write and sign. ** Increasing order of levels indicates improved socio-economic/village status.

**Table 2 nutrients-16-02872-t002:** Comparison of changes in WAZ-scores from baseline to 3 months and baseline to 60 months between intervention and control arms.

Age Category (Months)	Change in WAZ-Scores			Mean Difference (95% CI)
	Baseline—3 Months		*p*-Value ^1^	
	Intervention	Control		
06–11	0.08 ± 0.22 (−0.34, 0.51)	−0.44 ± 0.29 (−1.02, 0.15)	0.120	0.52 (−0.14, 1.18)
12–23	0.35 ± 0.11 (0.13, 0.57)	−0.18 ± 0.14 (−0.45, 0.09)	<0.001	0.53 (0.28, 0.78)
24–35	0.38 ± 0.15 (0.09, 0.66)	−0.24 ± 0.17 (−0.57, 0.09)	<0.001	0.62 (0.27, 0.95)
36–47	0.36 ± 0.14 (0.09, 0.63)	−0.13 ± 0.16 (−0.44, 0.18)	<0.001	0.49 (0.21, 0.77)
48–60	0.32 ± 0.22 (−0.12, 0.76)	−0.04 ± 0.24 (−0.52, 0.42)	0.031	0.37 (0.03, 0.71)
	Baseline—Last observation			
06–11	1.06 ± 0.21 (0.64, 1.48)	−0.65 ± 0.29 (−1.23, −0.08)	<0.001	1.71 (1.06, 2.36)
12–23	0.98 ± 0.12 (0.74, 1.22)	0.09 ± 0.15 (−0.19, 0.39)	<0.001	0.88 (0.61, 1.16)
24–35	0.75 ± 0.15 (0.45, 1.05)	−0.03 ± 0.18 (−0.38, 0.32)	<0.001	0.78 (0.44, 1.13)
36–47	0.71 ± 0.15 (0.41, 1.01)	0.21 ± 0.17 (−0.13, 0.54)	0.002	0.50 (0.19, 0.82)
48–60	0.30 ± 0.21 (−0.10, 0.71)	−0.20 ± 0.21 (−0.63, 0.23)	0.002	0.50 (0.20, 0.80)

Data shows mean ± standard error and 95% confidence interval for mean; ^1^ adjusted for age at enrolment as covariate and fixed factors group, gender, birth weight, parent literacy, wealth index, and village level status in both the models.

**Table 3 nutrients-16-02872-t003:** Association of factors with recovery from SUW at 3 and 60 months and recovery with relapse at 60 months—intervention arm.

Factors	Levels	Recovery at 3 Months (N = 335) ^†^		Recovery at 60 Months (N = 315) ^††^		Recovery with Relapse at 60 Months (N = 265)	
		Recovered/Total (%)	OR ^1^ [95% CI]; *p*-Value	Recovered/Total (%)	OR [95% CI]; *p*-Value	Recovery with Relapse/ Total Recovered (%)	OR ^2^ [95% CI]; *p*-Value
Age at enrolment (months)			1.01 [0.99–1.03]; 0.118		1.02 [0.99–1.04]; 0.193		0.99 [0.98–1.01]; 0.678
Sex	Female	62/159 (39.0)	Reference	119/148 (80.4)	Reference	55/119 (46.2)	Reference
	Male	63/177 (35.6)	0.86 [0.55–1.36]; 0.522	146/167 (87.4)	1.64 [0.87–3.09]; 0.128	72/146 (49.3)	1.09 [0.66–1.82]; 0.723
Birth weight (grams)	<2500	55/175 (31.4)	Reference	134/163 (82.2)	Reference	66/134 (49.3)	Reference
	≥2500	79/161 (43.5)	1.69 [1.07–2.68]; **0.025**	131/152 (86.2)	1.30 [0.69–2.48]; 0.417	61/131 (46.6)	0.86 [0.52–1.43]; 0.572
Parent literacy	Both illiterate	6/34 (25.9)	Reference	17/28 (60.7)	Reference	10/17 (58.8)	Reference
	Either or both literate	119/302 (39.4)	2.65 [1.04–6.79]; **0.042**	248/287 (86.4)	3.67 [1.52–8.85]; **0.004**	117/248 (47.2)	0.85 [0.29–2.43]; 0.760
Community	Nontribal	8/20 (40.0)	Reference	16/19 (84.2)	Reference	6/16 (37.5)	Reference
	Tribal	117/316 (37.0)	1.08 [0.39–2.99]; 0.877	249/296 (84.1)	1.04 [0.26–4.06]; 0.957	121/249 (48.6)	1.19 [0.39–3.68]; 0.758
Wealth index from assets (quartiles) *	I	33/96 (34.4)	Reference	78/89 (87.6)	Reference	40/78 (51.3)	Reference
	II	33/77 (42.9)	1.26 [0.66–2.39]; 0.490	63/72 (87.5)	0.89 [0.33–2.39]; 0.817	22/63(34.9)	0.68 [0.33–1.39]; 0.292
	III	30/84 (35.7)	1.05 [0.54–2.01]; 0.895	66/78 (84.6)	0.86 [0.33–2.22]; 0.750	35/66 (53.1)	1.42 [0.69–2.88]; 0.334
	IV	29/79 (36.7)	1.07 [0.55–2.10]; 0.835	58/76 (76.3)	0.50 [0.20–1.24]; 0.135	30/58 (51.7)	1.54 [0.73–3.27]; 0.256
Village level facilities (quartiles) *	I	41/127 (32.3)	Reference	102/119 (85.7)	Reference	62/102 (60.8)	Reference
	II	27/78 (34.6)	1.22 [0.64–2.31]; 0.546	59/72 (81.9)	0.95 [0.40–2.28]; 0.918	24/59 (40.7)	0.38 [0.19–0.78]; **0.008**
	III	18/38 (47.4)	1.70 [0.78–3.69]; 0.179	33/38 (86.8)	1.04 [0.33–3.26]; 0.947	10/33 (30.3)	0.26 [0.11–0.64]; **0.003**
	IV	39/93 (41.9)	1.61 [0.86–2.99]; 0.136	71/86 (82.6)	0.93 [0.39–2.20]; 0.860	31/71 (43.7)	0.46 [0.23–0.91]; **0.026**

^1^ Adjusted odds ratio using multiple logistic regression model; ^2^ adjusted odds ratio using multiple logistic regression treating recovery with relapse as an event and without relapse as a reference outcome; bold *p*-values indicate statistical significance. * Increasing levels indicate improved wealth/village status. ^†^ There were 3 deaths at the end of 3 months. ^††^ There were 4 deaths and 20 losses to follow-up cases until 60 months of observation for each child. Bold *p*-values indicate statistically significant values.

## Data Availability

Data collected for the study, including individual participant data and a data dictionary defining each field in the set, will not be made available to readers. As per the policy of Indian government, only anonymized/deidentified participant data will be made available upon request by editors or reviewers after due permission from the government of India. Related documents, such as informed consent form, study protocol, and statistical analysis plan, will be made available after publication if needed. These can be obtained from the corresponding author, Ashish Satav, drashish@mahantrust.org after due request.

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
