# Peer review of "Locally Prepared Therapeutic Food for Treatment of Severely Underweight Children in Rural India: An Interventional Prospective Controlled Community-Based Study with Long Follow-Up:—‘SAMMAN’ Trial"

_nutrients, 2024, doi:10.3390/nu16172872_

Round 1
Reviewer 1 Report
Comments and Suggestions for Authors
General comment
The authors may explain why they focus on SUW instead of the commonly used variable SAM.
Before a complete review of this paper can be done the authors should address the principal questions below and add the CONSORT-flow diagram.
Abstract:
Background
· The authors are surely aware of the WHO-guidelines for the Management of severe acute malnutrition from 2013 and 2023.
Objective
· Why 30% recovery? – Whilst the first objective is about treatment outcome, the second objective is on prevention of SAM. The authors may concentrate in this paper on treatment outcome only and prepare a second paper on prevention.
Setting
· Focus should be on the enrolled children in the intervention and the control group. What means ‘primary care for all participants’.
· The SAMMAN intervention was obviously directed to the children in the intervention group, what intervention experienced the children in the reference arm, e.g. standard treatment?
Results
· Add recovery rates of the children in the reference arm. How is the reduction of SUW prevalence in intervention villages related to the intervention?
Introduction:
Background
· ‘The prevalence of Severe Under Weight (S.U.W.) is 8·5% globally (9)’- This data is not based on the reference.
What was already known
· ‘S.U.W.’s prevalence is almost double that of Severe Acute Malnutrition (SAM).’ - - This data is not based on the reference.
The paragraph ‘Value addition of this study‘ should be presented under the discussion section,not in the introduction.
Methods
Study design, Setting and participants
· The authors may consider insert a CONSORT-Flow diagram in order to clarify the study flow. (https://www.equator-network.org/reporting-guidelines/consort/)
· If the authors compared the effect of the SUW-targeting intervention to non SUW-children it’s not what the title promised. A study of the effectivity of a ‘Local Therapeutic Food-Micro Nutrients’ for the management of SAM requires to compare this intervention to another intervention, e.g. RUTF, and children with the same degree of malnutrition.
...
Author Response
Author's Reply to the Review Report (Reviewer 1)
General Comment 1
The authors may explain why they focus on SUW instead of the commonly used variable SAM.
Reply from authors:
The joint statement published by U.N.I.C.E.F., W.H.O. and World Bank, focuses exclusively on wasting (W.H.Z. and M.U.A.C.) and stunting (H.A.Z.). The criteria used are mid–upper–arm–circumference (MUAC) and weight for height (WHZ) for malnutrition assessment and treatment, and not weight for age (WAZ). However, it does not consider underweight (weight for age: W.A.Z.). The prevalence of Severe Under Weight (SUW) is 9.8% (boys) and 7.3% (girls) in world,11% in India, >12% in rural India and 18·7% to 36.1% in tribal India. SUW has a high pooled mortality hazard ratio (H.R.) of 4·6 to 9·40. Thus, a large swath of severely underweight (SUW) is missed if WAZ is not used, eventually leading to the non-development of potential solutions for its prevention and management.
The WHO guidelines for the required daily allowance (RDA) of macro and micronutrients and duration of therapy for management of severe acute malnutrition are available, however there are no specific guidelines for SUW management. Their nutritional requirement, and length of therapy is unknown. The study by Islam highlights the importance of implementing present treatment guidelines of SAM for facility-based management of severely underweight children.
There is lot of literature available on effect of therapeutic food on SAM but scanty literature on effect of therapeutic food on SUW.
Hence, to find out solution to a major public health problem of SUW, we focussed on SUW. We will publish separate paper on SAM.
We have added in the main text.
Comment 2: Before a complete review of this paper can be done the authors should address the principal questions below and add the CONSORT-flow diagram.
Reply: We have addressed the principal questions below. We have also already added CONSORT-flow diagram as figure 1.
Abstract:
Background
Comment 3:
The authors are surely aware of the WHO-guidelines for the Management of severe acute malnutrition from 2013 and 2023.
Reply from authors: Thanks a lot for your comments. We are aware of the WHO-guidelines for the Management of SAM from 2013 and 2023. As there are no guidelines for management of SUW, we are trying to test the impact of LTF on SUW, so that new guidelines can be developed for SUW.
Objective:
Comment 4:
Why 30% recovery? - Whilst the first objective is about the treatment outcome, the second objective is on prevention of SAM. The authors may concentrate in this paper on treatment outcome only and prepare a second paper on prevention
Reply from authors:
The target of 30% recovery rate was achievable and significant from our past research conducted in similar settings. Hence, we kept 30% recovery as our primary objective.
We have included only treatments of SUW and did not include prevention of SAM as our objective.
We have conducted cluster randomized control trial to prevent SAM, which will be published as another paper.
We have added in the main text.
Setting:
Comment 5:
Focus should be on the enrolled children in the intervention and the control group. What means “primary care for all participants”.
Reply from authors: Our focus was on the enrolled children in the intervention and control group. The word ‘primary care for all participants’ was not included in our original manuscript. It was probably added by the editorial team of Nutrients. But usually, primary care indicates the basic management done for a patient at village level by paramedics. It also includes preventive measures.
Comment 6: The SAMMAN intervention was obviously directed to the children in the intervention group, what intervention experienced the children in the reference arm, e.g. standard treatment?)
Reply from authors:
In reference arm (RA), ‘Standard care”, consisting of BCC once in 10 weeks and treatment of infections as per WHO protocols, was provided. VHWs provided referral to government hospitals for the needy children. The children from RA who became SUW after enrollment were referred to hospitals.
After the SAMMAN Intervention of 90 days, “standard care”, was provided for children in both arms, till the age of 60 months.
We have added in the main text.
Results
Comment 7:
Add recovery rates of the children in the reference arm. How is the reduction of SUW prevalence in intervention villages related to the intervention?
Reply from authors:
The children in the reference arm were normal non -SUW children. Hence, we did not mention about their recovery.
The LTF-MN therapy of 90 days, intensive BCC and treatment of infections augmented the recovery of SUW and prevented the relapse. In intervention villages, treatment of infections and BCC of all children also prevented occurrence of new SUW cases. Intensive BCC changed the child rearing and child feeding, and hygienic practices of parents. It resulted in reduction of the prevalence of SUW.
We have added in the main text.
Introduction:
Background
Comment 8: ‘The prevalence of Severe Under Weight (S.U.W.) is 8·5% globally (9)’- This data is not based on the reference.
Reply from authors:
The prevalence of Severe Under Weight (S.U.W.) is 9.8% (boys) and 7.3% (girls) in world. We have corrected the reference.
The correct reference is : A D-N, Marrodán MD, A G, A V, Pacheco J, Sánchez-Álvarez M, et al. Female eco-stability and severe malnutrition in children: Evidence from humanitarian aid interventions of Action Against Hunger in African, Asian and Latin American countries. Nutricion Clinica y Dietetica Hospitalaria. 2017;34:127-34.
We have modified the reference in the main text.
What was already known
Comment 9: ‘S.U.W.’s prevalence is almost double that of Severe Acute Malnutrition (SAM).’ - - This data is not based on the reference.
Reply from authors:
Prevalence of S.U.W. is almost double that of Severe Acute Malnutrition (SAM). From our baseline study, the prevalence of severe underweight is 18.7% and prevalence of severe wasting is 7.1%. We have corrected the reference.
The correct reference is : Dani V, Satav A, Pendharkar J, Ughade S, Jain D, Adhav A, et al. Prevalence of under nutrition in under-five tribal children of Melghat: A community based cross sectional study in Central India. Clinical Epidemiology and Global Health. 2015;3(2):77-84.
The global prevalence of Severe Under Weight (S.U.W.) is 9.8% (boys) and 7.3% (girls), while the global prevalence of severe wasting was 3.9% for boys and 2.5% of girls.
The correct reference is : A D-N, Marrodán MD, A G, A V, Pacheco J, Sánchez-Álvarez M, et al. Female eco-stability and severe malnutrition in children: Evidence from humanitarian aid interventions of Action Against Hunger in African, Asian and Latin American countries. Nutricion Clinica y Dietetica Hospitalaria. 2017;34:127-34.
We have added in main text.
What was already known
Comment 10:
This paragraph ‘Value addition of this study” should be presented under the discussion section not in introduction section
Reply from authors:
We have Shifted the paragraph at the end of the discussion section.
Methods
Methods
Study design, Setting and participants
Comment 11:
The authors may consider insert a CONSORT flow diagram in order to clarify the study flow· (https://www.equator-network.org/reporting-guidelines/consort/)
Reply from authors: We have already included CONSORT flow diagram as figure 1.
Comment 12:
If the authors compared the effect of the SUW-targeting intervention to non SUW-children it’s not what the title promised. A study of the effectivity of a ‘Local Therapeutic Food-Micro Nutrients’ for the management of SAM requires to compare this intervention to another intervention, e.g. RUTF, and children with the same degree of malnutrition.
Reply from authors: Response: Thanks for the query. Please let us clarify why we chose the title. We concur that this was not a randomized controlled trial where we compared LTF-MN to RUTF. If we had the term “randomized controlled trial” would be in the title. Also we suspect that the moniker SAMAAN (Acronym for SAM- -Management) suggests that this trial was for treatment (Severe Acute Management). In actuality it was for treatment of SUW. We had a parallel trial for the treatment of SAM which will be reported separately, (for which the Moniker SAMAAN was used). Please see later response as to the derivation of the name SAMAAN).
Our title reflects what was done: Locally Prepared Therapeutic Food for Treatment of Severely Underweight Children in Rural India: An Interventional Prospective Controlled Community Based Study with Long Follow-up: - ‘SAMMAN’ Trial
We used LTF-MN to treat children with SUW in rural India, and while one course of treatment resulted in changes in WAZ, the seminal outcome of this trial was the long term impact of this one course of treatment on future growth of the children with SUW, up to adolescence. We show here that the growth of these children eventually was similar to that of control children without SUW. Hence the title completely reflects what was done. It was an interventional trial, data was collected prospectively for both arms of the study , the long term outcomes were compared to healthy controls in the community, which was the major focus of the study.
Comment 13; How is the reduction of SUW prevalence in intervention villages related to the intervention?
Reply from authors:
The LTF-MN therapy of 90 days, intensive BCC and treatment of infections augmented the recovery of SUW and prevented the relapse. In intervention villages, treatment of infections and BCC of all children also prevented occurrence of new SUW cases. Intensive BCC changed the child rearing and child feeding, and hygienic practices of parents. It resulted in reduction of the prevalence of SUW.
We have added in the main text.
Reviewer 2 Report
Comments and Suggestions for Authors
Local therapeutic food for severe underweight in India – Evaluation
· Strengths of the paper:
- The use of local ready-to-use or ready to cook therapeutic foods
- A prospective study with follow-up of children until early adolescence.
- Comparison group of age-matched children without underweight (at onset)
· Major weaknesses:
- No justification of the use of WAZ instead of WHZ to select and treat severely undernourished children
- Discussion lacking focus
- Too many figures and appendices that are not necessary, and too many acronyms
· Specific comments:
Abstract:
1. There ARE WHO guidelines for the management of severe malnutrition but no specific guidelines for the management of severe underweight
2. What is the meaning of Samman?
3. The cost should be given in US$.
4. It is no surprise that the change of WAZ is higher in I.A. children; these analyses and the statement thereupon are to be deleted from the abstract and from the main text.
Introduction:
5. It is too short. Previous work related to this study should be presented instead of the very short paragraph on ‘What is known’ The value added should go with the discussion or conclusion.
Methods:
6. What is the meaning of ‘representative’ study villages?
7. Better explain the purpose and the nature of the ‘’comparative’ arm children
8. The authors talk about intervention ‘villages’, but the intervention targeted severely underweight children: unclear.
9. The procedure for selected the intervention children is not clear. What happened to the SUW children who were not selected?
10. The village status is an interesting variable.
11. This and other parts of the paper need to be fully written; the bullet-type is not acceptable.
12. I do not believe that the paragraph on PPI is needed.
13. The ready-to-eat or ready-to-heat local therapeutic foods should be briefly described in the main text.
Results and discussion:
14. As mentioned above, comparing WAZ changes between IA and CA is not appropriate.
15. P. 5, line 1: Unclear sentence: ‘Sustained positive impact on growth pattern of SUW till early adolescence’
16. P. 5: Data on SUW prevalence in other studies should not appear here
17. The CFR was higher in the comparison children and among the severely underweight: were the ‘control’ children who became severely underweight treated? There is one ambiguous sentence on this while this is an important ethics point.
18. The tables and figures (the essential ones) should be better described and discussed.
19. The discussion has to be better organized and go into more depth, notably for the relapse, the age-related differences, the interest of local RUTF, etc.
20. A conclusion is needed.
Tables, figures and appendices
21. There are too many. Only the appendices useful for an international audience should be retained.
22. Table 3: It is suggested to use bold font for significant differences so that the table is easier to read.
23. Table 4 should definitely be excluded, although some points can be discussed, but this study did not compare the local with the international RUTFs.
24. In our view, figures 5, 6, 7 and 9 could be removed.
25. The appendices are interesting, but belong to a project report more that to a scientific paper. Those that could be retained are 1, 6, 7 and 11. The essential points from the other appendices could be integrated into the main text.
Author Response
Author's Reply to the Review Report (Reviewer 2)
Reviewer 2
Strengths of the paper
The use of local ready-to-use or ready to cook therapeutic foods
- A prospective study with follow-up of children until early adolescence.
- Comparison group of age-matched children without underweight (at onset)
Reply from authors: Thanks a lot for your kind and warm words.
Major weaknesses:
- No justification of the use of WAZ instead of WHZ to select and treat severely undernourished children
Reply from authors:
The joint statement published by U.N.I.C.E.F., W.H.O. and World Bank, focuses exclusively on wasting (W.H.Z. and M.U.A.C.) and stunting (H.A.Z.). The criteria used are mid–upper–arm–circumference (MUAC) and weight for height (WHZ) for malnutrition assessment and treatment, and not weight for age (WAZ). However, it does not consider underweight (weight for age: W.A.Z.). The prevalence of Severe Under Weight (SUW) is 9.8% (boys) and 7.3% (girls) in world,11% in India, >12% in rural India and 18·7% to 36.1% in tribal India. SUW has a high pooled mortality hazard ratio (H.R.) of 4·6 to 9·40. Thus, a large swath of severely underweight (SUW) is missed if WAZ is not used, eventually leading to the non-development of potential solutions for its prevention and management.
The WHO guidelines for the required daily allowance (RDA) of macro and micronutrients and duration of therapy for management of severe acute malnutrition (WHZ) are available, however there are no specific guidelines for SUW management. Their nutritional requirement, and length of therapy is unknown. The study by Islam highlights the importance of implementing present treatment guidelines of SAM for facility-based management of severely underweight children.
There is lot of literature available on effect of therapeutic food on WHZ but scanty literature on effect of therapeutic food on WAZ.
Hence, to find out solution to a major public health problem of SUW (WAZ<-3 SD), we focussed on WAZ. We have also conducted research to find out impact of LTF-MN on WHZ <-3 SD. We will publish separate paper on SAM.
We have added in the main text.
- Comment: Discussion lacking focus
Reply from authors: Thanks a lot for the good suggestion. We have corrected the discussion in the main text and made it focussed.
- Comment: Too many figures and appendices that are not necessary, and too many acronyms
Reply from authors: We have reduced the figures, appendices and the acronyms.
Abstract
Comment 1. There ARE WHO guidelines for the management of severe malnutrition but no specific guidelines for the management of severe underweight
Reply from authors:
The WHO guidelines, specifically, for the management of 'Severe Acute malnutrition' (SAM) , in the form of required daily allowance (RDA) for macro-nutrients,, micro-nutrients and duration of nutritional therapy are available. However, similar guidelines for the management of severe underweight (SUW), which is a form of 'acute on chronic malnutrition', are not available. The prevalence of SUW is almost double to that of SAM and has very high mortality hazard ratio.
Such lack of specific guidelines for the SUW management, was impetus to use the available WHO guidelines of SAM management, for SUW management in this study, from 2011-2023. Our LTF (100 gm )-MN (5gm) provides 13.5-16 grams of proteins, 450-550 calories and 20-30 gms of fats along with 40 micronutrients as per WHO UNICEF guidelines. The government of India has started a similar program in 2023.
We have added in the main text.
Comment:2 What is the meaning of Samman?
Reply from authors:
SAMMAN is the acronym for SAM Management. The word SAMMAN in the Hindi language means respect. Thus, we are treating the severely undernourished children with respect.
We have added in the main text.
Comment 3 The cost should be given in US$.
Reply from authors:
The cost of management per SUW child was INR 3888 (US$47) less than RUTF.
We have added in the main text.
Comment 4. It is no surprise that the change of WAZ is higher in I.A. children; these analyses and the statement thereupon are to be deleted from the abstract and from the main text.
Reply from authors: Thanks a lot for your suggestion. But all the authors including the biostatistician and the epidemiologist consider ‘that the WAZ is higher in I.A. Children, the analyses and the statement there upon’ is important. It is humble request from us that this statement should not to be deleted from the abstract or the main text. But editor has to take final decision.
Introduction:
Comment: 5. It is too short. Previous work related to this study should be presented instead of the very short paragraph on ‘What is known’ The value added should go with the discussion or conclusion.
Reply from authors: We have modified the introduction. We have shifted the value added to the last part of the discussion.
Introduction:
Malnutrition contributes to 45 – 68.2% of under-five mortality and for survivors, has long‐term implications for educational achievement, economic productivity, and the risk of non‐communicable diseases in later life. The Millenium Development Goals had set a target of reducing the prevalence of malnutrition to half between 1990 and 2015 and the Sustainable Development Goals (SDG) target 2·2, is to end all forms of malnutrition by 2030. The UNICEF, WHO, World Bank Group Joint Malnutrition estimates focus exclusively on obesity, stunting (HAZ) and wasting prevalence. They use mid-upper arm circumference and weight for height criteria for malnutrition assessment and treatment and did not use weight for age (WAZ) to define or treat malnutrition. Moderate to severe underweight for age babies (MUW and SUW), usually precedes these more severe manifestations of malnutrition. Use of the weight for height or height for age anthropometric criteria, misses a large swath of SUW, eventually leading to the lack of the development of potential solutions for its prevention and management. The prevalence of SUW defined as more than three standard deviations (SD) below the median WAZ is 9.8% (boys) and 7.3% (girls) in world, 11% in India, overall and 12·1% to 18·7% in rural India and 18·7% to 36.1% in tribal India. The pooled mortality hazard ratio (HR) of SUW is high, 4·6 to 9·40.
The WHO guidelines for the required daily allowance (RDA) of macro and micronutrients and duration of therapy for management of severe acute malnutrition are available, however there are no specific guidelines for SUW management. Their nutritional requirement, and length of therapy is unknown. The study by Islam highlights the importance of implementing present treatment guidelines of SAM for facility-based management of severely underweight children.
What was already known:
PubMed, Cochrane, Google-Scholar, and B.M.J. databases were searched for global and Indian studies, using the following terms- SUW, community–based management of severe malnutrition, ready to use therapeutic food (R.U.T.F.), LTF, etc. till March 24.
Prevalence of SUW is almost double that of Severe Acute Malnutrition (SAM). From our baseline study, the prevalence of severe underweight is 18.7% and prevalence of severe wasting is 7.1%. While the global prevalence of severe wasting was 3.9% for boys and 2.5% of girls, and that of Severe Under Weight (SUW) was 9.8% (boys) and 7.3% (girls).
In almost all previous studies, feeding is unsupervised and supplementary in nature. The Ready to use therapeutic food (R.U.T.F.) packets/ ration are distributed as takeaways to be fed at home. However, this has not significantly impacted SUW recovery/mortality. The significant burden of SUW and lack of successful trials to guide the community-based management of SUW were the impetus for this trial.
‘MAHAN’ is providing hospital and community-based health and nutrition services to the rural–tribal community of Melghat, India, since 1998.
Recognizing this lacuna, we initiated this trial. This study aims to assess the impact of community-based management of SUW using locally prepared, culturally acceptable, affordable, palatable multiple therapeutic foods with micronutrients on SUW children.
Primary objective is to achieve a recovery rate of around 30% at the end of 90 days and to assess recovery at the age of 60 months in severely underweight (SUW) children between 6 to 60 age, in a population of 14000 from 14 tribal villages of Melghat, central India. The target of 30% recovery rate was achievable and significant from our past research conducted in similar settings.(1) The secondary objectives are: a)To compare the change in WAZ score and growth pattern of intervention arm and reference arm (normal age and time-matched) up to the age of 5 years in above setting. b) To compare the Case Fatality Rate (CFR) of SUW children in the Intervention Arm (IA), with the non-SUW in Reference Arm (RA), at three stages: at the end of 90 days of treatment, at 60 months of age and at early adolescence (9–13 years). c) To reduce point prevalence of SUW by at least 35% at the age of 5 years (2019) as compared to baseline (2011) in intervention villages. The exploratory objective was to assess the impact of SUW management on SUW relapse rate up to the age of 60 months.
Methods:
Comment : 6. What is the meaning of ‘representative’ study villages?
Reply from authors:
Melghat, a rural -tribal/ area in Maharashtra, India, consisting of 320 villages(clusters), divided in Dharni and Chikhaldara blocks with a population of approximately 3,00,000, of which 84 % are tribals. The Dharni block was divided in five zones, after stratification, based on distance from base hospital, three clusters from each zone were selected. One cluster dropped out due to unwillingness for participation. Thus 14 representative study villages were randomly selected (Appendix 1). These villages represent the demographic, socioeconomic characteristics, health indicators, and malnutrition (SUW) of maximum Melghat villages. Hence, they are representative study villages of Melghat.
We have added in the main text.
Comment 7: Better explain the purpose and the nature of the ‘’comparative’ arm children
Reply from authors:
The purpose of adding comparative reference arm (RA) of normal children was to know whether the impact of our intervention on recovery, growth pattern, BMI, and case fatality rate of SUW children is similar to, non-SUW normal children at the age of 60 months. So that the treated SUW children in IA would have similar outcome indicators to those of normal children in RA. Our ethical committee did not permit, diagnosing SUW child and yet not treating them, as that would be unethical. Hence, we included normal, non-SUW children as RA.
We have added in the main text.
Comment 8 The authors talk about intervention ‘villages’, but the intervention targeted severely underweight children: unclear.
Reply from authors:
We conducted our intervention in this study at the village level and therefore those villages are named as intervention villages. In the intervention villages, we targeted severely underweight children (SUW) as intervention arm (IA) for treatment by LTF-MN to know its impact. In the same intervention villages, we selected reference arm (R.A.)- non-SUW children, age-matched (±two weeks) as controls.
We have added in the main text.
Comment 9: The procedure for selected the intervention children is not clear. What happened to the SUW children who were not selected?)
Reply from authors:
Enrollment: In every month, VHWs conducted a door-to-door survey and recorded the anthropometry of all children in the villages between 6 and 60 months of age using a complete enumeration method, standardized digital weighing machines, and stadiometer/infantometer. Our team of experts selected all SUW from the anthropometry data. Our trained medical supervisors cross checked anthropometry of all SUW and SAM children, other supervisors cross checked anthropometry of MUW and MAM. The final list of SUW children was prepared for enrollment for interventions. Thus, during enrolment, we included all the SUW children present in the villages.
We have added in the main text.
Comment 10 The village status is an interesting variable.
Reply from authors: Thanks for your comment.
Comment 11 This and other parts of the paper need to be fully written; the bullet-type is not acceptable.
Reply from authors: We have fully revised this and other parts of the paper. We have removed the bullet-type. Thanks for your suggestions.
Comment 12 : I do not believe that the paragraph on PPI is needed.
Reply from authors: Thanks to reviewer for the suggestion. But all authors are of opinion that PPI was an important component of the success of the trial. Without PPI, it was not possible to achieve the targets. Hence, the paragraph is essential for readers and others who want to replicate such programs. Hence it is humble request to keep the paragraph on PPI.
Comment (13) : The ready-to-eat or ready-to-heat local therapeutic foods should be briefly described in the main text.
Reply from authors:
Our LTF consisted of 8 different varieties of culturally acceptable ready- to- eat and ready- to- heat feeds. The ready- to- eat feeds like ‘chivda’ and ‘chikki’ were given as breakfast and evening snacks and ready-to-heat feeds like ‘khichadi’,’thalipeeth’ and ‘upama’ were given for lunch and dinner. From 1998 to 2010 we conducted diet survey, multiple interviews and focus group discussions to understand the dietary pattern and cultural acceptability of food by the local community especially the children. We designed our LTF as per the dietary pattern and cultural acceptability of local children.
We also used the WHO guidelines, specifically, for the management of 'Severe Acute malnutrition' (SAM), in the form of required daily allowance (RDA) for macro-nutrients, micro-nutrients while designing LTF.
Our LTF (100 gm)-MN (5gm) provides 13.5-16 grams of proteins, 450-550 calories and 20-30 gms of fats along with 40 micronutrients as per WHO UNICEF guidelines.
We have added in the main text.
Results and discussion:
Comment 14: As mentioned above, comparing WAZ changes between IA and CA is not appropriate.
Reply from authors: Thanks to reviewer for the suggestion. One reviewer mentioned it as strength of the paper.
But all the authors (epidemiologist, biostatistician, pediatricians, nutrition, local experts) are of the opinion that comparing WAZ changes between intervention arm and control arm is essential to understand the impact of our intervention on recovery, growth pattern, BMI, and case fatality rate of SUW children in intervention arm ( IA) with non-SUW normal children (reference arm). Our ethical committee did not permit handling of SUW child without treating it and hence we included non SUW children as reference arm.
Comment 15: P. 5, line 1: Unclear sentence: ‘Sustained positive impact on growth pattern of SUW till early adolescence’)
Reply from authors:
Follow–up was done till early adolescence (October 2023), to understand the long-term impact on survival and anthropometry. All children were successfully tracked for survival. 65 children from IA and 142 children from CA went to other villages; hence anthropometry could not be done. There was no difference in CFR in both groups (1·17%). The difference of mean BMI and proportion of mortality between two groups was not significant (p=0·952 and p=0·999 respectively). The probability density of BMI followed normal distribution in both the groups as revealed in Figure 8. The follow up of treated SUW children revealed that they achieved growth pattern and BMI similar to normal adolescents, thus, indicating a sustained positive impact of our intervention This sustained impact is possible due to intensive BCC, which positively changed the child rearing, feeding, and hygienic practices of parents; and timely and appropriate community-based treatment of infections.
We have added in the main text.
Comments 16: P. 5: Data on SUW prevalence in other studies should not appear here.
Reply from authors:
Government of India accepts the reduction in prevalence as a major criterion to evaluate the impact of any intervention. Hence, we studied the point prevalence of SUW in intervention villages.
Other reviewers asked to discuss for mixed results in literature for the prevalence of SUW in intervention and control arms. Hence, we compared it with other studies.
We have modified the text.
Comment 17: The CFR was higher in the comparison children and among the severely underweight: were the ‘control’ children who became severely underweight treated? There is one ambiguous sentence on this while this is an important ethics point.
Reply from authors:
The children in reference or control arm who were initially normal but eventually became SUW were offered standard of care and treatment through Government facilities for malnutrition management. We got permission from ethical committee for the same. So, there is no ethical issue.
We have added in the main text.
Comment 18: The tables and figures (the essential ones) should be better described and discussed.
Reply from authors: We have described and discussed the essential tables and figures.
Comment 19: The discussion has to be better organized and go into more depth, notably for the relapse, the age-related differences, the interest of local RUTF, etc.
Reply from authors: We have modified the discussion in the main text. We have added the relapse, age related differences and interest of local RUTF into more depth as written below.
- Recovery with / without relapse and associated factors:
The recovery without relapse was treated as a reference outcome, while recovery
with relapse was considered as the outcome of interest. Out of the 265 recovered children,138 (40·7%) had no relapse and were normal at the end of study indicating importance of our intensive BCC and treatment of infections till the age of 5 years. Younger children, in the age group of 6–24 months, had significantly more relapses i.e. 61% (81) compared to 35% (46) in the 25–60 months age group (p <0·0001). During early age, the immunological system is week as compared to older children and hence more prone for infection. During the growth trajectory, we also noticed the episodes of infection (pneumonia, diarrhoea, tuberculosis) which preceded the relapses. Similarly, the chewing capacity of smaller children is less as compared to older children, hence the relapses are more in early age group. The mature rotatory chewing movements of molar teeth and backward pushing of food by tongue comes during the age of 24 to 30 months which help for proper intake and digestion of food in older children. BCC should focus on cooking of separate special food for children below the age of 24 months by parents, after completion of LTF therapy.
Significant risk factors for relapses were scarce communication facilities, difficult to access health-care and the absence of safe drinking water. Comparison between recovery at 60 months with and without relapse showed, that the likelihood of recovery without relapse increases significantly as the village–level facilities improved.
Few studies have treated SUW, without focusing on relapse, default rate, infections, and recovery.
- The Interest of using LTF-MN
The cost of SUW child management was INR 3888/child (US$47) for 90 days therapy of LTF–MN, which is cheaper than RUTF of Maharashtra Government, INR 9438/child (US$113) and ComPAS study (standard protocol US$1041). Systematic review by Fetriyuna has revealed that local therapeutic food are more cost effective than standard RUTF.
The low cost of LTF-MN used in our intervention is attributed using locally available food produce, preparations by local manpower with the use of minimal technology. On the other hand, RUTF is industrially produced with higher marketing cost. LTF-MN being cheaper, we could provide LTF to a large number of SUW children and SAM children from 2011-2023.
Nutrition programs with LTF-MN therapy can be implemented on larger scale, without increasing economic burden by Government and other organizations, which will help to achieve the SDG Goal 2.2.
LTF is perceived as food prepared by local tribal females and with BCC it helped in improved cooking practices in community even after therapy. RUTF is considered as industrial and medicinal sachets for sick children and not as food. It will not improve cooking practices in community. Hence people using RUTF lose confidence in home food for SUW treatment.
LTF is prepared locally, creating employment for tribal females. It is prepared in socio-culturally accepted way, as variety of preparations, adding diversity and palatability of food overcoming the limiting factors. LTF builds confidence in the local community that severe malnutrition can be treated with their own food. Hence acceptance rate is >93%.
RUTF is single dish and hence acceptance rate is very less. (40%)
Systematic review by Fetriyuna has revealed that local therapeutic foods are more acceptable than standard RUTF.
In most studies RUTF packets for 1 to 2 weeks are given to parents, to be consumed at home by severely malnourished child, however sharing with other members of family cannot be controlled affecting the recovery of index case. In the present study all LTF-MN feeds given three to four times a day, are directly supervised by VHW at community feeding centre, in clean and hygienic surrounding. This has improved our recovery rate. Hence therapeutic food should be given to SAM, SUW in nutrition feeding centers, not at home of the beneficiaries. MAHAN LTF-MN has the better recovery rate for SUW as compared to RUTF (<30%).
Case fatality rate (CFR) after RUTF is 1 to 5%, even when, most of these studies have treated uncomplicated SUW. In our study, CFR is 0.29% for SUW who were treated with LTF even when, we have treated both complicated and uncomplicated SUW.
Our prior study of LTF-MN therapy for SAM management had recovery rate of 79.4%.
In the present study, analysis of SAM and severe stunting out of treated SUW in IA, show significant improvement in the recovery and reduction in proportion of SAM (p < 0·0001) and severe stunting (p < 0·0001), indicating impact of SAMMAN intervention on most serious cases of combined SAM and SUW, and severe stunting and SUW. This highlights that recovery of stunting is possible with LTF-MN, during long term follow-up coupled with regular BCC and treatment of infections. Hence LTF is more sustainable and replicable as compared to RUTF.
Comment 20: A conclusion is needed.
Reply from authors:
This study, with good recovery, few adverse effects, culturally acceptable, safe and cost-effective with locally available resources, can be a sustainable and replicable model for other impoverished, difficult to access settings globally. Nutrition programs should include W.A.Z. <–3 as an additional independent criterion for malnutrition therapy.
We have added the conclusion in the main text.
Tables, figures and appendices
Comment 21: There are too many. Only the appendices useful for an international audience should be retained.
Reply from authors: The authors agree with reviewers and reduced the figures, tables and appendices as deemed appropriate by the reviewers.
Comment number 22: Table 3: It is suggested to use bold font for significant differences so that the table is easier to read.
Reply from authors: We have edited the table 3 as per suggestions.
Table 3. Association of factors with Recovery from SUW at 3 and 60 months and Recovery with relapse at 60 months – Intervention arm
Factors |
Levels |
Recovery at 3 months (N=335)† |
Recovery at 60 months (N=315) †† |
|
Recovery with relapse at 60 months (N=265) |
|
|
Recovered/Total (%) |
OR1 [95% CI]; P-value |
Recovered/Total (%) |
OR[95% CI]; P-value |
Recovery with relapse/ Total recovered (%) |
OR2 [95% CI]; P-value |
||
Age at enrolment (months) |
1·01 [0.99-1.03]; 0.118 |
|
1·02 [0·99-1·04]; 0·193 |
|
0·99 [0·98-1·01]; 0·678 |
||
Sex |
Female |
62/159 (39·0) |
Reference |
119/148 (80·4) |
Reference |
55/119 (46·2) |
Reference |
|
Male |
63/177 (35·6) |
0·86 [0·55-1·36]; 0·522 |
146/167 (87·4) |
1·64 [0·87-3·09]; 0·128 |
72/146 (49·3) |
1·09 [0·66-1·82]; 0·723 |
Birth weight (grams) |
< 2500 |
55/175 (31·4) |
Reference |
134/163 (82·2) |
Reference |
66/134 (49·3) |
Reference |
³ 2500 |
79/161 (43·5) |
1·69 [1·07-2·68]; 0·025 |
131/152 (86·2) |
1·30 [0·69-2·48]; 0·417 |
61/131 (46·6) |
0·86 [0·52-1·43]; 0·572 |
|
Parent literacy |
Both illiterate |
6/34 (25·9) |
Reference |
17/28 (60·7) |
Reference |
10/17 (58·8) |
Reference |
|
Either or both literate |
119/302 (39·4) |
2·65 [1·04-6·79]; 0·042 |
248/287 (86·4) |
3·67 [1·52-8·85]; 0·004 |
117/248 (47·2) |
0·85 [0·29-2·43]; 0·760 |
Community |
Non-tribal |
8/20 (40·0) |
Reference |
16/19 (84·2) |
Reference |
6/16 (37·5) |
Reference |
|
Tribal |
117/316 (37·0) |
1·08 [0·39-2·99]; 0·877 |
249/296 (84·1) |
1·04 [0·26-4·06]; 0·957 |
121/249 (48·6) |
1·19 [0·39-3·68]; 0·758 |
Wealth index from assets (quartiles)* |
I |
33/96 (34·4) |
Reference |
78/89 (87·6) |
Reference |
40/78 (51·3) |
Reference |
II |
33/77 (42·9) |
1·26 [0·66-2·39]; 0·490 |
63/72 (87·5) |
0·89 [0·33-2·39]; 0·817 |
22/63(34·9) |
0·68 [0·33-1·39]; 0·292 |
|
|
III |
30/84 (35·7) |
1·05 [0·54-2·01]; 0·895 |
66/78 (84·6) |
0·86 [0·33-2·22]; 0·750 |
35/66 (53·1) |
1·42 [0·69-2·88]; 0·334 |
|
IV |
29/79 (36·7) |
1·07 [0·55-2·10]; 0·835 |
58/76 (76·3) |
0·50 [0·20-1·24]; 0·135 |
30/58 (51·7) |
1·54 [0·73-3·27]; 0·256 |
Village level facilities (quartiles)* |
I |
41/127 (32·3) |
Reference |
102/119 (85·7) |
Reference |
62/102 (60·8) |
Reference |
II |
27/78 (34·6) |
1·22 [0·64-2·31]; 0·546 |
59/72 (81·9) |
0·95 [0·40-2·28]; 0·918 |
24/59 (40·7) |
0·38 [0·19-0·78]; 0·008 |
|
|
III |
18/38 (47·4) |
1·70 [0·78-3·69]; 0·179 |
33/38 (86·8) |
1·04 [0·33-3·26]; 0·947 |
10/33 (30·3) |
0·26 [0·11-0·64]; 0·003 |
|
IV |
39/93 (41·9) |
1·61 [0·86-2·99]; 0·136 |
71/86 (82·6) |
0·93 [0·39-2·20]; 0·860 |
31/71 (43·7) |
0·46 [0·23-0·91]; 0·026 |
1Adjusted odds ratio using multiple logistic regression model; 2Adjusted odds ratio using multiple logistic regression treating recovery with relapse as an event and without relapse as reference outcome; Bold p-values indicate statistical significance . *Increasing levels indicates improved wealth/village status; †There were 3 deaths at the end of 3 months. ††There were 4 deaths and 20 loss to follow-up cases till 60 months of observation for each child.
Comment 23: Table 4 should definitely be excluded, although some points can be discussed, but this study did not compare the local with the international RUTFs.
Reply from authors: Thanks a lot for good suggestion. We have removed the table 4 from main article and shifted to appendix as supplementary material Supplementary Table 4 (Table S4). As per reviewer’s suggestion, we have included the important points of table 4 in the discussion. It will be useful for readers especially policy makers and institutes who want to replicate LTF.
- The LTF-MN and RUTF
The cost of SUW child management was INR 3888/child (US$47) for 90 days therapy of LTF–MN, which is cheaper than RUTF of Maharashtra Government, INR 9438/child (US$113) and ComPAS study (standard protocol US$1041). Systematic review by Fetriyuna has revealed that local therapeutic food is more cost effective than standard RUTF.
The low cost of LTF-MN used in our intervention is attributed using locally available food produce, preparations by local manpower with the use of minimal technology. On the other hand, RUTF is industrially produced with higher marketing cost. LTF-MN being cheaper, we could provide LTF to a large number of SUW children and SAM children from 2011-2023.
Nutrition programs with LTF-MN therapy can be implemented on larger scale, without increasing economic burden by Government and other organizations, which will help to achieve the SDG Goal 2.2.
LTF is perceived as food prepared by local tribal females and with BCC it helped in improved cooking practices in community even after therapy. RUTF is considered as industrial and medicinal sachets for sick children and not as food. It will not improve cooking practices in community. Hence people using RUTF lose confidence in home food for SUW treatment.
LTF is prepared locally, creating employment for tribal females. It is prepared in socio-culturally accepted way, as variety of preparations, adding diversity and palatability of food overcoming the limiting factors. LTF builds confidence in the local community that severe malnutrition can be treated with their own food. Hence acceptance rate is >93%.
RUTF is single dish and hence acceptance rate is very less. (40%)
Systematic review by Fetriyuna has revealed that local therapeutic foods are more acceptable than standard RUTF.
In most studies RUTF packets for 1 to 2 weeks are given to parents, to be consumed at home by severely malnourished child, however sharing with other members of family cannot be controlled affecting the recovery of index case. In the present study all LTF-MN feeds given three to four times a day, are directly supervised by VHW at community feeding centre, in clean and hygienic surrounding. This has improved our recovery rate. Hence therapeutic food should be given to SAM, SUW in nutrition feeding centers, not at home of the beneficiaries. MAHAN LTF-MN has the better recovery rate for SUW as compared to RUTF (<30%).
Case fatality rate (CFR) after RUTF is 1 to 5%, even when, most of these studies have treated uncomplicated SUW. In our study, CFR is 0.29% for SUW who were treated with LTF even when, we have treated both complicated and uncomplicated SUW.
Our prior study of LTF-MN therapy for SAM management had recovery rate of 79.4%.
In the present study, analysis of SAM and severe stunting out of treated SUW in IA, show significant improvement in the recovery and reduction in proportion of SAM (p < 0·0001) and severe stunting (p < 0·0001), indicating impact of SAMMAN intervention on most serious cases of combined SAM and SUW, and severe stunting and SUW. This highlights that recovery of stunting is possible with LTF-MN, during long term follow-up coupled with regular BCC and treatment of infections. Hence LTF is more sustainable and replicable as compared to RUTF.
Comment 24: In our view, figures 5, 6, 7 and 9 could be removed.
Reply from authors: Thanks a lot for the suggestions. Other reviewer has asked to describe the points mentioned in figure 5 i.e. prevalence of SUW. It is also important for Government of India’s policy which always focus on the impact of any intervention, on prevalence of severe malnutrition. Hence, we have kept it. We have shifted old figure 6 and 7 as supplementary figure 1 and 2 as it will be useful for policymakers. So now total figures have been reduced to 7.
Comment 25: The appendices are interesting, but belong to a project report more that to a scientific paper. Those that could be retained are 1, 6, 7 and 11. The essential points from the other appendices could be integrated into the main text.
Reply from authors: Thanks to reviewers for good suggestions. We have removed few of the appendices and kept appendices 1, 6, 7 and 11. We have prepared a new supplementary material file.
Adding the essential points from other appendices may cross the word limit of manuscript. If editors allow it, we will add some essential points from other appendices into the main text of manuscript.
Reviewer 3 Report
Comments and Suggestions for Authors
This is a very important topic. I commend the authors for bringing together the study that is well designed, thorough, and findings well-articulated. I have some minor comments below.
P 16, Paragraph 2, please elaborate more for clarity
The novelty of the study with regards to long-term follow up should be discussed earlier in the Discussion.
Results of BMI could be briefly discussed in results
‘The proportion of children with S.U.W. in I. A (14·7%) is lesser than R.A. (19·2%)’- reasons for possible increase in SUW in RA should be discussed
Need to discuss reasons for mixed results in literature for the prevalence of SUW in IA and RA.
Discussion section refers to studies ‘Bangladesh hospital’, ‘Malawi study’, without giving much details. Please provide some details and context to compare.
Please discuss more the implications of reduced cost LTF to treat SUW. This can also be used to treat stunting and wasting, micronutrient deficiencies and towards achieving the SDG Goal 2.2.
Author Response
Author's Reply to the Review Report (Reviewer 3)
REVIEWER 3
Comments and Suggestions for Authors:
Comment 1: This is a very important topic. I commend the authors for bringing together the study that is well designed, thorough, and findings well-articulated. I have some minor comments below.
Reply from authors: Thanks a lot for your warm and kind encouraging words. We have tried to address the comments as below.
Comment 2: P 16, Paragraph 2, please elaborate more for clarity
Reply from authors: There is conflict of interest statement on page 16 (of previous version of manuscript), paragraph 2. Shall we elaborate it or some other paragraph? Please clarify.
Comment 3: The novelty of the study with regards to long-term follow up should be discussed earlier in the Discussion.
Reply from authors: Thanks for your comments. We did follow up of children after studying their recovery, growth pattern, case fatality, prevalence and relapses. Hence, we discussed it in the later part of the manuscript. But if the reviewer and editor think, it should be discussed earlier, we will shift it.
Comment 4: Results of BMI could be briefly discussed in results
Reply from authors:
Result section: Follow–up was done till early adolescence (October 2023), to understand the long-term impact on survival and anthropometry. All children were successfully tracked for survival. 65 children from IA and 142 children from CA went to other villages; hence anthropometry could not be done. There was no difference in CFR in both groups (1·17%). The difference of mean BMI and proportion of mortality between two groups was not significant (p=0·952 and p=0·999 respectively). The probability density of BMI followed normal distribution in both the groups as revealed in Figure 8. The follow up of treated SUW children revealed that they achieved growth pattern and BMI similar to normal adolescents, thus, indicating a sustained positive impact of our intervention
Discussion section: The BMI of IA and RA children were almost the same up–to early adolescence indicating that the recovered SUW children follow a growth pattern similar to normal children. This sustained impact is possible due to intensive BCC, which positively changed the child rearing, feeding, and hygienic practices of parents; and timely and appropriate community-based treatment of infections.
We have added the results of BMI and its discussion in main text.
Comment 5: ‘The proportion of children with S.U.W. in I. A (14·7%) is lesser than R.A. (19·2%)’- reasons for possible increase in SUW in RA should be discussed
Reply from authors:
In the natural course of growth, some of the children who were normal in the beginning in RA became SUW due to common infections like pneumonia, diarrhea, malaria and tuberculosis. Other factors were poverty and chronic scarcity of nutrition.
In the IA, due to LTF-MN, treatment of infections and BCC, the proportion of children with SUW was less than RA at the age of 60 months.
We have added in the main text.
Comment 6: Need to discuss reasons for mixed results in literature for the prevalence of SUW in IA and RA.
Reply from authors:
The significant reduction in the point prevalence of SUW in intervention villages (p<0.001) (Figure 5) indicates the sustained impact of SAMMAN intervention. The LTF-MN therapy of 90 days, intensive BCC and treatment of infections augmented the recovery of SUW and prevented the relapse. In intervention villages, treatment of infections and BCC of all children also prevented occurrence of new SUW cases. Intensive BCC changed the child rearing and child feeding, and hygienic practices of parents. It resulted in reduction of the prevalence of SUW.
Mixed results in literature are as follows:
Impact of intensive BCC without supplementary feeding:
In Palghar study, the prevalence of SUW was reduced from 32.9% to 23.9% (p value =0.002) due to intensive BCC for 18 months by health staff without food supplementation. In Udupi study, prevalence of SUW reduced significantly in intervention area (from 8.69% to 3.16%) as compared to control area because of health education and nutrition demonstration. In our previous study of tribal area, the prevalence of severe malnutrition was reduced significantly, net reduction by 50.52% in intervention villages over control villages (p<0.01) with intensive BCC and community-based treatment of infections without supplementary feeding.
Supplementary feeding with BCC:
In tribal Rajasthan study, the supplementary feedings containing total 500-700 calories per day were given as three supervised feeds in government Nutrition Care Centers and two feeds at home. Follow up was done for 6 months. There was no intensive BCC and treatment of infections. Hence the SUW prevalence in Rajasthan study was reduced from 32.9% to 26.1%.
Supplementary feeding without BCC:
In Bangladesh study, supervised supplementary feeding (8-9 gms proteins, 300 kcal) was given to severely underweight children for 4 months , with no significant difference in the prevalence in the project and non-project areas (11.4% vs 12.1%) because of improper and inadequate supplementary food which replaced home food, inadequate counselling of parents, reduced or no appetite of sick SUW, and sharing of food with other siblings.
We have added in the main text.
Comment 7: Discussion section refers to studies ‘Bangladesh hospital’, ‘Malawi study’, without giving much details. Please provide some details and context to compare.
Reply from authors:
In the Rural Malawi RCT study, corn-soya blend or lipid-based nutrient were monthly distributed for 12 weeks at home for SAM children of 6-18 months age. Their secondary data analysis revealed insignificant recovery of SUW (p=0.211) because of the short duration of unsupervised supplementary feedings, no intensive BCC and the lack of monitoring for sharing of food with other family members.
In the Bangladesh Integrated Nutrition Project, supplementary food was given to severely underweight children in community. The recovery of SUW was not significant, nearly zero impact on WAZ. The possible causes are mistargeting, sharing of supplementary food; substitution for other foods, inadequate food, incomplete participation, culturally less palatable food, no intensive BCC, no capacity building activities and inadequate feeding practices.
We have added in the main text.
Comment 8:
Please discuss more the implications of reduced cost LTF to treat SUW. This can also be used to treat stunting and wasting, micronutrient deficiencies and towards achieving the SDG Goal 2.2.
Reply from authors: Thank you very much for your comment. The implications of reduced cost of LTF to treat SUW, stunting, wasting, micronutrient deficiencies are as follows:
The cost of SUW child management was INR 3888/child (US$47) for 90 days therapy of LTF–MN, which is cheaper than RUTF of Maharashtra Government, INR 9438/child (US$113) and ComPAS study (standard protocol US$1041). Systematic review by Fetriyuna has revealed that local therapeutic food are more cost effective than standard RUTF.
The low cost of LTF-MN used in our intervention is attributed using locally available food produce, preparations by local manpower with the use of minimal technology. On the other hand, RUTF is industrially produced with higher marketing cost. LTF-MN being cheaper, we could provide LTF to a large number of SUW children and SAM children from 2011-2023.
Nutrition programs with LTF-MN therapy can be implemented on larger scale, without increasing economic burden by Government and other organizations, which will help to achieve the SDG Goal 2.2.
LTF is perceived as food prepared by local tribal females and with BCC it helped in improved cooking practices in community even after therapy.
LTF is prepared locally, creating employment for tribal females. It is prepared in socio-culturally accepted way, as variety of preparations, adding diversity and palatability of food overcoming the limiting factors. LTF builds confidence in the local community that severe malnutrition can be treated with their own food. Hence acceptance rate is >93%.
In the present study all LTF-MN feeds given three to four times a day, are directly supervised by VHW at community feeding centre, in clean and hygienic surrounding. This has improved our recovery rate. Hence therapeutic food should be given to SAM, SUW in nutrition feeding centers, not at home of the beneficiaries. MAHAN LTF-MN has the better recovery rate for SUW as compared to RUTF (<30%).
Our prior study of LTF-MN therapy for SAM management had recovery rate of 79.4%.
In the present study, analysis of SAM and severe stunting out of treated SUW in IA, show significant improvement in the recovery and reduction in proportion of SAM (p < 0·0001) and severe stunting (p < 0·0001), indicating impact of SAMMAN intervention on most serious cases of combined SAM and SUW, and severe stunting and SUW. This highlights that recovery of stunting is possible with LTF-MN, during long term follow-up coupled with regular BCC and treatment of infections. Hence LTF is more sustainable and replicable.
We have added in the main text.
Round 2
Reviewer 2 Report
Comments and Suggestions for Authors
This is an important study. It is unfortunate that the presentation is not very good. The revised version is better but there are still some questions and concerns:
1. The authors should answer the following question: What is the interest of using WAZ instead of WHZ or HAZ as malnutrition (and recovery) criterion? This is important for the relevance of the study.
2. Introduction: 'Use of WH or HA misses a large swath of SUW': please add at least one reference.
3. Methods: How was the follow-up of SUW children conducted up to adolescence?
4. Please add subtitles in the discussion (there is only one presently).
5. p.5: Breasfeeding only 'allowed'? Not promoted?
6. p.8, last sentence: Greater than in what group?
7. p.10: Under 'Recovery with or without relapse: These data should be presented in the Results section.
8. P.10, 'Chewing capacity': Please add relevant reference.
9. Figure 7 is unclear.
10. Text to delete:
- Introduction: P.2, last sentence
- P.3, last paragraph: (Whether) the impact of our intervention on recovery
- P.6: The demographic characteristics were summarized.
- P.7: a) b) c) d)
- P.13: Value addition paragraph: not needed.
11. RDA: RECOMMENDED (Not required) DIETARY (not daily) allowances
12. Keywords: Please add 'India'.
Comments on the Quality of English LanguagePlease edit the newly added text in particular.
Author Response
Author’s response: We have provided point-by-point responses to the reviewer’s comments
Reviewer no: 2
Comment 1
- The authors should answer the following question: What is the interest of using WAZ instead of WHZ or HAZ as malnutrition (and recovery) criterion? This is important for the relevance of the study.
Reply from authors:
Re: Using WAZ for the study group that we studied.
We have previously responded to this question. We will briefly summarise what we have pointed out previously.
Severe underweight (SUW) infants and children were the chosen group to study rather than severe acute malnutrition (SAM) or stunted children. SUW is defined using WAZ, whereas SAM is defined using WHZ and stunting uses HAZ. Since we chose to study SUW we had to use WAZ for both defining the study population as well as using it as the primary outcome measure for the acute outcomes.
We chose to study SUW because
- The prevalence of SUW in our population and globally is more than twice that of SAM. Our published study (2016) has revealed the prevalence of SAM as 7.1% and Severe Underweight (SUW) 18.7%. The study by UNICEF (2018) has revealed the prevalence of SAM and SUW in tribal areas of India as 5.66% and 20.27% respectively. While the global prevalence of SAM was 3.9% for boys and 2.5% of girls, and that of SUW was 9.8% (boys) and 7.3% (girls) (2017).
Use of the weight for height anthropometric criteria, misses a large swath of SUW, eventually leading to the lack of the development of potential solutions for its prevention and management.
- SUW preceded SAM often and is a milder form of malnutrition
- Despite this, there is a significant mortality in children with SUW. The pooled mortality hazard ratio (HR) of SUW is high, 4·6 to 9·40.
- There are numerous randomised controlled trials of SAM and stunting but none that specifically looked at SUW
- The WHO guidelines for the Recommended Dietary Allowance (RDA) of macro and micronutrients and duration of therapy for management of SAM are available, however there are no specific guidelines from WHO or UNICEF for SUW management despite it being a significant problem. Their nutritional requirement, and length of therapy is unknown.
- If SUW preceded SAM and there is a higher prevalence of severe underweight, then logically this is the place and the group to direct nutritional interventions.
- Yet there are no randomised controlled trials of SUW in this population, despite many studies of SAM some of which secondarily analysed data for patients with SUW.
- Thus we chose to study SUW, because it is a major problem, it precedes other forms of severe malnutrition, of itself there is a significant mortality, there are no randomised trials of interventions in this population, and consequently there are no accepted guidelines from WHO and UNICEF for this population.
Why we chose to use WAZ as the primary outcome measure:
- Given that the inclusion criteria used WAZ for defining the study population, this is clearly the acute on chronic outcome measure that needs to be studied as a primary outcome and followed throughout the follow-up period.
- However, we clearly also used WHZ and HAZ for determining the long-term outcomes of the acute intervention of SUW with BCC, the hypothesis being does 90 days of treatment of SUW and sustained BCC have a long-term impact on the proportion of children that develop SAM or stunting, and is this different from the rates in controls. Please see Figure 7 for this outcome as well as the results and discussion sections for the implications of this intervention.
- The Sustainable Development Goals (SDG) target 2·2 , is to end all forms of malnutrition by 2030, The UNICEF, WHO, World Bank Group joint Malnutrition estimates focus on prevalence of wasting (WHZ), stunting(HAZ) and obesity, and stress on prevention of stunting. They use mid-upper arm circumference and weight for height (WHZ) criteria for malnutrition assessment and treatment5 and did not use weight for age (WAZ) to define or treat malnutrition. Severe underweight ( SUW ) (acute on chronic malnutrition), is defined as WAZ less than three standard deviations (SD) below the median.
- The study by Islam highlights the importance of implementing present treatment guidelines of SAM for facility-based management of severely underweight children.
We have added summary of above explanation in the main text. If reviewer wants and editor permits, we will be happy to write the above detail reply in the main manuscript.
Comment 2: Introduction: 'Use of WH or HA misses a large swath of SUW': please add at least one reference.
Reply from authors : There are references for “Use of the weight for height anthropometric criteria, misses a large swath of SUW”.
The references are as follows:
Prevalence of Severe Under Weight (SUW) (WAZ <-3SD) is almost double that of Severe Acute Malnutrition (SAM) (WHZ <-3SD). The prevalence of severe underweight is 18.7% and prevalence of SAM (WHZ <-3SD) is 7.1%. The reference is : Dani V, Satav A, Pendharkar J, Ughade S, Jain D, Adhav A, et al. Prevalence of under nutrition in under-five tribal children of Melghat: A community based cross sectional study in Central India. Clinical Epidemiology and Global Health. 2015;3(2):77-84.
The global prevalence of SUW is 9.8% (boys) and 7.3% (girls), while the global prevalence of SAM(WHZ <-3SD) was 3.9% for boys and 2.5% of girls. The reference is : A D-N, Marrodán MD, A G, A V, Pacheco J, Sánchez-Álvarez M, et al. Female eco-stability and severe malnutrition in children: Evidence from humanitarian aid interventions of Action Against Hunger in African, Asian and Latin American countries. Nutricion Clinica y Dietetica Hospitalaria. 2017;34:127-34.
Clusters of world |
Prevalence of SAM (WHZ <-3SD) (%) (Male, Female) |
Prevalence of SUW (WAZ <-3SD) (%) (Male, Female) |
Central America |
0.9, 0.6 |
6, 4.9 |
Central Asia |
3.7, 2.2 |
16.6, 15.2 |
South coast of Asia |
4.1, 2.6 |
15.4, 13.8 |
Sahel |
5.7, 3.7 |
9.2, 6.4 |
West coast of Africa |
2.6, 2.0 |
9.1, 7.4 |
Central Africa |
2.3, 1.4 |
11.4, 8.2 |
East Africa |
2.7, 1.7 |
6.8, 6.1 |
All countries |
3.9, 2.5 |
9.8, 7.3 |
The study by UNICEF (2018) has revealed the prevalence of SAM and SUW in various tribal areas of India.
Areas of India |
Prevalence of SAM (WHZ <-3SD) (%) |
Prevalence of SUW (WAZ <-3SD) (%) |
Khuntpani (JH) |
8.10 |
36.10 |
Nabarangpur (OD) |
7.30 |
23.50 |
Koraput (OD) |
4.10 |
18.20 |
Naraini (UP) |
6.30 |
21.50 |
Kesla (MP) |
3.80 |
11.10 |
Total (2299 children) |
5.66 |
20.27 |
We have added a summary of above data with references in the main text.
We have removed these words “height for age” from the above comment in main text.
Comment 3:
Methods: How was the follow-up of SUW children conducted up to adolescence?
Reply from authors: Thanks a lot for your comments. Our village health workers (VHWs) recorded anthropometry of all study participants in the villages, every month, till the age of 60 months (2011-2019). In 2019, the age group of study participants was from 4 to 9 years. All study participants were annually followed up during home visits by VHWs till the adolescent age. There after anthropometry of all the study participants was recorded once after 5 years i.e. in 2024 by VHWs. The age group of study participants in 2024 was 9 to 14 years i.e. adolescence age. All the data was verified by supervisors.
We have added it in the main text.
Comment 4:
- Please add subtitles in the discussion (there is only one presently).
Reply from authors: Thanks a lot for good suggestions. Subtitles have been added in the discussion section of main manuscript. These are marked by underlines.
Comment 5:
p.5: Breastfeeding only 'allowed'? Not promoted?
Reply from authors: Breastfeeding was allowed and promoted.
We have added in the main text.
Comment 6: p.8, last sentence: Greater than in what group?
Reply from authors: In Hossain’s study in Bangladesh, the rates of weight gain were greater in SUW group who received cereal based supplementary food (p< 0.05) as compared to other groups who did not receive supplementary food.
We have added in the main text.
Comment 7:
p.10: Under 'Recovery with or without relapse: These data should be presented in the Results section.
Reply from authors: Recovery with / without relapse and associated factors:
The recovery without relapse was treated as a reference outcome, while recovery
with relapse was considered as the outcome of interest. Out of the 265 recovered children,138 (40·7%) had no relapse and were normal at the end of study. Younger children, in the age group of 6–24 months, had significantly more relapses i.e. 61% (81) compared to 35% (46) in the 25–60 months age group (p <0·0001).
Significant risk factors for relapses were scarce communication facilities, difficult to access health-care and the absence of safe drinking water. Comparison between recovery at 60 months with and without relapse showed, that the likelihood of recovery without relapse increases significantly as the village–level facilities improved.
We have added in main text.
Comment 8: P.10, 'Chewing capacity': Please add relevant reference.
Reply from authors : We have added following references in the main text.
There is an increase in masticatory performance (Chewing capacity )of children as they grow up.
Mastication, or chewing, is a learned task that involves a complex masticatory system and many muscles. The development of mastication is influenced by dental eruption and the development of oro-facial structures. Here are some milestones in the development of mastication:
4–6 months: Jaw movements are simple elevations, assisted by the tongue and lips. 12 months: The basic pattern of jaw muscle activity for mastication is established. 18 months: A rotary motion, which is a hallmark of mature mastication, may be seen. 24–30 months: Lateral jaw motion emerges, and eventually transitions to a rotary jaw movement. 10–14 years: A transition to adult-type masticatory behaviour occurs.
Mastication is a highly coordinated task that involves seven of the twelve cranial nerves, as well as some cervical nerves. The muscles of mastication include the temporalis, masseter, medial pterygoid, and lateral pterygoid. Cyclical, rotary jaw movement that involves food transport to the teeth by the tongue, followed by a quick closing of the jaw to connect the teeth with the bolus, and, lastly, a power stroke, where the jaw ultimately closes against the resistance of the bolus. (Meenakshi & Paul, 2017).
Comment 9: Figure 7 is unclear.
Reply from authors: The alluvial plot displaying transition of SUW children as regards severe stunting and severe wasting from baseline to 60 months of age. The recovery in severe wasting and severe stunting was observed in 86.7% and 45.3% cases respectively at 60 months of age, as per Figure 7.
We have added above text in results section of manuscript.
Comment 10: Text to delete:
- Introduction: P.2, last sentence
Reply from authors: We have deleted it from last sentence of page 2. Authors consider it as important sentence. Hence, we have made change, mixed it with next paragraph and removed the duplication. The new sentence is as follows.
We initiated this trial, because of the significant burden of SUW and lack of guidelines for the community-based management of SUW.
- - P.3, last paragraph: (Whether) the impact of our intervention on recovery.
Reply from authors: The above sentence was added as reply to comments by one reviewer and academic editor during first round of review.
- - P.6: The demographic characteristics were summarized.
Reply from authors: We have deleted the above statement.
- - P.7: a) b) c) d)
Reply from authors: We have deleted the above a) b) c) d) from P.7.
- - P.13: Value addition paragraph: not needed.
Reply from authors: Value addition paragraph was added at the end of discussion as per suggestions of 2 different reviewers during round one.
Comment 11: RDA: RECOMMENDED (Not required) DIETARY (not daily) allowances.
Reply from authors: We have removed the required daily allowances words and replaced it with Recommended Dietary Allowances. Thanks for the suggestion.
Comment 12: Keywords: Please add 'India'.
Reply from authors: We have added India in keywords.
We hope that our revised manuscript will be found acceptable for publication in Nutrients.
Sincerely yours
Ashish R Satav (MBBS, MD)
President,
MAHAN trust, India
Corresponding author:
Ashish R Satav (MBBS, MD)
President, MAHAN trust, Melghat, Maharashtra, India
C/O Mahatma Gandhi Tribal Hospital, Karmgram, Utavali, Dharni
District: Amravati, State: Maharashtra, India. Pin code: 444 702
Phone: Mobile: +919423118877, Tele-fax: +91712723989
Email: drashish@mahantrust.org
